# When Expressivity Meets Trainability: Fewer than $n$ Neurons Can Work

**Jiawei Zhang**[*]
Shenzhen Research Institute of Big Data
The Chinese University of Hong Kong,
Shenzhen, China
jiaweizhang2@link.cuhk.edu.cn

**Yushun Zhang**[*]
Shenzhen Research Institute of Big Data
The Chinese University of Hong Kong,
Shenzhen, China
yushunzhang@link.cuhk.edu.cn

**Mingyi Hong**
University of Minnesota - Twin Citie
MN, USA
mhong@umn.edu

**Ruoyu Sun** [†]
University of Illinois at Urbana-Champaign
IL, USA
ruoyus@illinois.edu

**Zhi-Quan Luo**
Shenzhen Research Institute of Big Data
The Chinese University of Hong Kong,
Shenzhen, China
luozq@cuhk.edu.cn

## Abstract

Modern neural networks are often quite wide, causing large memory and computation costs. It is thus of great interest to train a narrower network. However, training narrow neural nets remains a challenging task. We ask two theoretical questions: Can narrow networks have as strong expressivity as wide ones? If so, does the loss function exhibit a benign optimization landscape? In this work, we provide partially affirmative answers to both questions for 1-hidden-layer networks with fewer than $n$ (sample size) neurons when the activation is smooth. First, we prove that as long as the width $m \geq 2n/d$ (where $d$ is the input dimension), its expressivity is strong, i.e., there exists at least one global minimizer with zero training loss. Second, we identify a nice local region with no local-min or saddle points. Nevertheless, it is not clear whether gradient descent can stay in this nice region. Third, we consider a constrained optimization formulation where the feasible region is the nice local region, and prove that every KKT point is a nearly global minimizer. It is expected that projected gradient methods converge to KKT points under mild technical conditions, but we leave the rigorous convergence analysis to future work. Thorough numerical results show that projected gradient methods on this constrained formulation significantly outperform SGD for training narrow neural nets.

## 1 Introduction

Modern neural networks are huge (e.g. [8, 74]). Reducing the size of neural nets is appealing for many reasons: first, small networks are more suitable for embedded systems and portable devices; second, using smaller networks can reduce power consumption, contributing to "green computing".

---

[*]Equal contribution. These authors are listed in alphabetical order.
[†]Corresponding author: Ruoyu Sun.

35th Conference on Neural Information Processing Systems (NeurIPS 2021).

There are many ways to reduce network size, such as quantization, sparcification and reducing the width (e.g. [20, 77]). In this work, we focus on reducing the width (training narrow nets).

Reducing the network width often leads to significantly worse performance [3]. What is the possible cause? From the theoretical perspective, there are three possible causes: worse generalization power, worse trainability (how effective a network can be optimized), and weaker expressivity (how complex the function a network can represent; see Definition 1). Our simulation shows that the training error deteriorates significantly as the width shrinks, which implies that the trainability and/or expressivity are important causes of the worse performance (see Section 5.1 for more evidence ). We do not discuss generalization power for now, and leave it to future work.

So how is the training error related to expressivity and trainability? The training error is the sum of two parts (see, e.g., [64]): the expressive error (which is the best a given network can do; also the global minimal error) and the optimization error (which is the gap between training error and the global minimal error; occurs because the algorithm may not find global-min). The two errors are of different nature, and thus need to be discussed separately.

It is understandable that narrower networks might have weaker expressive power. What about optimization? There is also evidence that smaller width causes optimization difficulty. A number of recent works show that increasing the width of neural networks helps create a benign empirical loss landscape ([19, 35, 59]), while narrow networks (width $m <$ sample size $n$) suffer from bad landscape ([2, 60, 66, 72, 79]). Therefore, if we want to improve the performance of narrow networks, it is likely that both expressiveness and trainability need to be improved.

The above discussion leads to the following two questions:

> **(Q1)** *Can a narrow network have as strong* **expressivity** *as a wide one?*
>
> **(Q2)** *If so, can a local search method find a (near) globally optimal solution?*

The key challenges in answering these questions are listed below:

- It is not clear whether a narrow network has strong expressivity or not. Many existing works focus on verifying the relationship between zero-training-error solutions and stationary points, but they neglect the (non)existence of such solutions (e.g. [71], [63]). For narrow networks, the (non)-existence of zero-training-error-solution is not clear.

- Even if zero-training-error solutions do exist, it is still not clear how to reach those solutions because the landscape of a narrow neural network can be highly non-convex.

- Even assuming that we can identify a region that contains zero-training-error solutions and has a good landscape, it is potentially difficult to keep the iterates inside such a good region. One may think of imposing an explicit constraint, but this approach might introduce bad local minimizers on the boundary [6].

In this work, we (partially) answer **(Q1)** and **(Q2)** for a 1-hidden-layer nets with fewer than $n$ neurons. Our main contributions are as follows:

- **Expressiveness and nice local landscape.** We prove that, as long as the width $m$ is larger than $2n/d$ (where $n$ is the sample size and $d$ is the input dimension), then the expressivity of the 1-hidden-layer net is strong, i.e., w.p.1. there *exists* at least one global-min with zero empirical loss. In addition, such a solution is surrounded by a good local landscape with no local-min or saddles. Note that our results do not exclude the possibility that there are sub-optimal local minimizers on the *global* landscape.

- **Every KKT point is an approximated global minimizer.** For the original unconstrained optimization problem, the nice *local* landscape does not guarantee the *global* statement of "every stationary point is a global minimizer". We propose a constrained optimization problem that restricts the hidden weights to be close to the identified nice region. We show

---

[3]This can be verified on our empirical studies in Section 5. Another evidence is that structure pruning (reducing the number of channels in convolutional neural nets (CNN)) is known to achieve worse performance than unstructured pruning; this is an undesirable situation since many practitioners prefer structure pruning (due to hardware reasons).

that every Karush–Kuhn–Tucker (KKT) point is an approximated global minimizer of the unconstrained training problem [4].

- In real-data experiments, our proposed training regime can significantly outperforms SGD for training narrow networks. We also perform ablation studies to show that the new elements proposed in our method are useful.

## 2 Background and Related Works

The expressivity of neural networks has been a popular topic in machine learning for decades. There are two lines of works: One focuses on the *infinite-sample* expressivity, showing what functions of the entire domain can and cannot be represented by certain classes of neural networks (e.g. [4, 45]). Another line of works characterize the *finite-sample* expressivity, i.e. how many parameters are required to memorize finite samples (e.g. [5, 13, 21, 25, 26, 42, 75]). The term "expressivity" in this work means the finite-sample expressivity; see Definition 1 in Section 4.1. A major research question regarding expressivity in the area is to show *deep* neural networks have much stronger expressivity than the *shallow* ones (e.g. [7, 17, 41, 48, 56, 57, 67, 70, 72]). However, all these works neglect the trainability.

In the finite-sample case, wide networks (width poly($n$)) have both strong representation power (i.e. the globally minimal training error is zero) and strong trainability (e.g. for wide enough nets, Gradient Descent (GD) converges to global minima [1, 16, 28, 80]). While these wide networks are often called "over-parameterized", we notice that the number of parameters of a width-$n$ network is actually at least $nd$, which is much larger than $n$. If comparing $n$ with the number of parameters (instead of neurons), the transition from under-parameterization and over-parameterization for a one-hidden-layer fully-connected net (FCN) does not occur at width-$n$, but at width-$n/d$. In this work, we will analyze networks with width in the range $[n/d, n)$, which we call "narrow networks" (though rigorously speaking, we shall call them "narrow but still overparameterized networks").

There are a few works on the trainability of narrow nets (one-hidden-layer networks with $m \geq n/d$ neurons). Soudry and Carmon [63], Xie et al. [71] show that for such networks, stationary points with full-rank NTK (neural tangent kernel) are zero-loss global minima. However, it is not clear whether the NTK stays full rank during the training trajectory. In addition, these two works do not discuss whether a zero-loss global minimizer exists.

There are two interesting related works [9, 12] pointed out by the reviewers. Bubeck et al. [9] study how many neurons are required for memorizing a finite dataset by 1-hidden-layer networks. They prove the following results. Their first result is an "existence" result: there exists a network with width $m \geq \frac{4n}{d}$ which can memorize $n$ input-label pairs (their Proposition 4). However, in this setting they did not provide an algorithm to find the zero-loss solution. Their second result is related to algorithms: they proposed a training algorithm that achieves accuracy up to error $\epsilon$ for a neural net with width $m \geq O\left(\frac{n}{d} \frac{\log(1/\epsilon)}{\epsilon}\right)$. This result requires width dependent on the precision $\epsilon$; for instance, when the desired accuracy $\epsilon = 1/n$, the required width is at least $O\left(\frac{n^2}{d}\right)$. In contrast, in our work, the required number of neurons is just $2n/d$, which is independent of $\epsilon$.

Daniely [12] also studies the expressivity and trainability of 1-hidden-layer networks. To memorize $n(1 - \epsilon)$ random data points via SGD, their required width is $\tilde{O}(n/d)$. They assumed $n = d^c$ where $c > 0$ is a *fixed* constant (appeared in Sec. 3.3 of [12]), in which case the hidden factor in $\tilde{O}$ is $O(\log[d(\log d)]^c)$. In other words, if $n, d \to \infty$ with the scaling $n = d^c$ for a fixed constant $c$, then their bound is roughly $O(n/d)$ up to a log-factor. Nevertheless, for more general scaling of $n, d$, the exponent $c = (\log n)/(\log d)$ may not be a constant and the hidden factor may not be a log-factor (e.g. for fixed $d$ and $n \to \infty$). We tracked their proof and find that the width bound for general $n, d$ is $O\left((n/d)(\log(d \log n))^{\log n/\log d}\right)$, which can be larger than $O\left(n^2/d\right)$ (see detailed computation and explanation in Appendix C). In contrast, our required width is $2n/d$ for arbitrary $n$ and $d$. Our bound is always smaller than $n$ when $d > 2$.

---

[4]This result describes the loss landscape of the constrained optimization problem, not directly related to algorithm convergence. Nevertheless, it is expected that first-order methods converge to KKT points and thus approximate global minimizers. A rigorous convergence analysis may require verifying extra technical conditions, which is left to future work.

Additionally, there is a major difference between Daniely [12] and our work: they analyze the original unconstrained problem and SGD; in contrast, we analyze a constrained problem. This may be the reason why we can get a stronger bound on width. In the experiments in Section 5, we observe that SGD performs badly when the width is small (see the 1st column in Figure 4 (b)). Therefore, we suspect an algorithmic change is needed to train narrow nets with such width (due to the training difficulty), and we indeed propose a new method to train narrow nets.

Due to the space constraints, we defer more related works in Appendix B.

## 3 Challenges For Analyzing Narrow Nets

In this section, we discuss why it is challenging to achieve expressivity and trainability together for narrow nets. Consider a dataset $\{(x_i, y_i)\}_{i=1}^n \subset \mathbb{R}^d \times \mathbb{R}$ and a 1-hidden-layer neural network:

$$f(x; \theta) = \sum_{j=1}^m v_j \sigma \left( w_j^T x \right), \tag{1}$$

where $\sigma \left( w_j^T x \right)$ is the output of the $j$-th hidden nodes with hidden weights $w_j$, $\sigma(\cdot)$ is the activation function, and $v_j$ is the corresponding outer weight (bias terms are ignored for simplicity). To learn such a neural network, we search for the optimal parameter $\theta = (w, v)$ by minimizing the following empirical (training) loss:

$$\min_\theta \ell(\theta) = \frac{1}{2} \sum_{i=1}^n \left( y_i - f\left( x_i; \theta \right) \right)^2, \tag{2}$$

The gradient of the above problem w.r.t. hidden weights $w = \{w_i\}_{i=1}^m$ is given by: $\nabla_w \ell(\theta) = J(w; v)^T (f(w; v) - y) \in \mathbb{R}^{md \times 1}$, where $J(w; v) \in \mathbb{R}^{n \times md}$ is the Jacobian matrix w.r.t $w$:

$$J(w; v) := \begin{bmatrix} \nabla_w f\left( w; x_1, v \right)^T \\ \vdots \\ \nabla_w f\left( w; x_n, v \right)^T \end{bmatrix} = \begin{bmatrix} v_1 \sigma'\left( w_1^T x_1 \right) x_1^T & \cdots & v_m \sigma'\left( w_m^T x_1 \right) x_1^T \\ & \vdots & \\ v_1 \sigma'\left( w_1^T x_n \right) x_n^T & \cdots & v_m \sigma'\left( w_m^T x_n \right) x_n^T \end{bmatrix} \in \mathbb{R}^{n \times md}. \tag{3}$$

First order methods like GD converge to a stationary point $\theta^* = (w^*, v^*)$ (i.e. with zero gradient) under mild conditions [6]. For problem (2), it is easy to show that if (i) $(w^*, v^*)$ is a stationary point, (ii) $J(w^*; v^*) \in \mathbb{R}^{n \times md}$ is full row rank and $n \leq md$, then $(w^*, v^*)$ is a global-min (this claim can be proved by setting the partial gradient of (2) over $w$ to be zero). In other words, for training a network with width $m \geq n/d$, an important tool is to ensure the full-rankness of the Jacobian.

Recent works have shown that it is possible to guarantee the full rankness of the Jacobian matrix along the training trajectories, however, the required width is above $\Omega(\text{poly}(n))$. Roughly speaking, the proof sketch is the following: (i) with high probability, $J(w; v)$ is non-singular locally around the random initialization ([71], [63]), (ii) increasing the width can effectively bound the parameter movement from initialization, so the "nice" property of non-singualr $J(w, v)$ holds throughout the training, leading to a linear convergence rate [16]. Under this general framework, a number of convergence results are developed for wide networks with width $\Omega(\text{poly}(n))$ ([1, 11, 28, 34, 49, 53–55, 80, 81]). This idea is also illustrated in Figure 1 (a).

We notice that there is a huge gap between the necessary condition $m \geq n/d$ and the common condition $m \geq \Omega(\text{poly}(n))$. We suspect that it is possible to train a narrow net with width $\Theta(n/d)$ to small loss. To achieve this goal, we need to understand why existing arguments require a large width and cannot apply to a network with width $\Theta(n/d)$.

The first reason is about trainability. The above arguments no longer hold when the width is not large enough to control the movement of hidden weights. In this case, the iterates may easily travel far away from the initial point and get stuck at some singular-Jacobian critical points with high training loss (see Figure 1 (b). Also see Figure 4 (b) & (d) for more empirical evidence). In other words, GD may get stuck at sub-optimal stationary points for narrow nets.

The second reason, and also an easily ignored one, is the expressivity (a.k.a. the representation power, see Definition 1 for a formal statement). In above discussion, we implicitly assumed that there exists a zero-loss global minimizer, which is equivalent to "there exists a network configuration such that the network can memorize the data". For networks with width at least $n$, this assumption can be

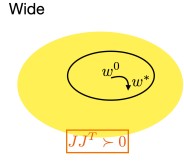
Wide

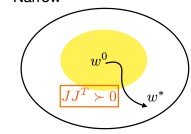
Narrow

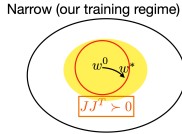
Narrow (our training regime)

(a) Wide networks  (b) Narrow networks  (c) Narrow networks (our regime)

Figure 1: The parameter movement under different regimes. The shaded area indicates the region where $J(w; v)$ is non-singular, the black circle denotes the region that GD iterates will explore, and the red circle is the constraint designed in our training regime, it will be discussed in Section 4.3.

justified in the following way. The feature matrix

$$\Phi(w) := \begin{bmatrix} \sigma\left(w_1^T x_1\right), \ldots, \sigma\left(w_m^T x_1\right) \\ \vdots \\ \sigma\left(w_1^T x_n\right), \ldots, \sigma\left(w_m^T x_n\right) \end{bmatrix} \in \mathbb{R}^{n \times m} \tag{4}$$

can span the whole space $\mathbb{R}^n$ when it is full rank and $m \geq n$, thus the network can perfectly fit any label $y \in \mathbb{R}^n$ even without training the hidden layer. It is important to note that when the width $m$ is below the sample size $n$, full row-rankness does not ensure that the row space of the feature matrix is the whole space $\mathbb{R}^n$. In other words, it is not clear whether a global-min with zero loss exists.

In the next section, we will describe how we obtain strong expressive power with $\Theta(n/d)$ neurons, and how to avoid sub-optimal stationary points.

## 4 Main Results

### 4.1 Problem Settings and Preliminaries

We denote $\{(x_i, y_i)\}_{i=1}^n \subset \mathbb{R}^d \times \mathbb{R}$ as the training samples, where $x_i \in \mathbb{R}^d$, $y_i \in \mathbb{R}$. For theoretical analysis, we focus on 1-hidden-layer neural networks $f(x; \theta) = \sum_{j=1}^m v_j \sigma\left(w_j^T x\right) \in \mathbb{R}$, where $\sigma(\cdot)$ is the activation function, $w_j \in \mathbb{R}^d$ and $v_j \in \mathbb{R}$ are the parameters to be trained. Note that we only consider the case where $f(x; \theta) \in \mathbb{R}$ has 1-dimensional output for notation simplicity.

To learn such a neural network, we search for the optimal parameter $\theta = (w, v)$ by minimizing the empirical loss (2), and sometimes we also use $\ell(w; v)$ or $f(w; x, v)$ to emphasize the role of $w$. We use the following shorthanded notations: $x := (x_1^T; \ldots; x_n^T) \in \mathbb{R}^{n \times d}$, $y := (y_1, \ldots, y_n)^T \in \mathbb{R}^n$, $w := (w_1, \ldots, w_m)_{j=1}^m \in \mathbb{R}^{d \times m}$, $v := (v_1, \ldots, v_m)_{j=1}^m \in \mathbb{R}^m$, and $f(w; v) := (f(x_1; w, v), f(x_2; w, v), \ldots, f(x_n; w, v))^T \in \mathbb{R}^n$. We denote the Jacobian matrix of $f(w; v)$ w.r.t $w$ as $J(w; v)$, which can be seen in (3). We define the feature matrix $\Phi(w)$ as in (4). We denote the operator $\nabla_w$ as "taking the gradient w.r.t. $w$", and the same goes for $\nabla_v$. Throughout the paper, 'w.p.1' is the abbreviation for 'with probability one'; when we say 'in the neighborhood of initialization', it means '$w$ is in the neighborhood of the initialization $w^0$'.

Now, we formally define the term "expressivity". As discussed in Section 2, we focus on the finite-sample (as opposed to infinite-sample) expressivity, which is relevant in practical training.

**Definition 1.** *(Expressivity) We say a neural net function class $\mathcal{F} = \{f(x; \theta); \theta \in \Theta\}$ has strong ($n$-sample) expressivity if for any $n$ input-output pairs $D = \{(x_i, y_i)\}_{i=1}^n \subset \mathbb{R}^d \times \mathbb{R}$ where $x_i$'s are distinct, there exists a $\hat{\theta}(D) \in \Theta$ such that $f(x_i; \hat{\theta}(D)) = y_i$, $i = 1, \cdots, n$. Or equivalently, the optimal value of empirical loss (2) equals 0 for any $D$. Sometimes we may drop the word "strong" for brevity.*

Next, let us describe the *mirrored LeCun's* initialization in Algorithm 1. The idea is that through this initialization, the hidden outputs will cancel out with the outer weights, so that we get zero initial output for any input $x$; see Figure 2 for a simple illustration. Note that similar symmetric initialization strategies are also proposed in some recent works such as [11] and [12]. However, our purpose is different. More explanation can be seen in the final paragraph of Section 4.2.

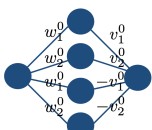

Figure 2: A simple example of the mirrored LeCun's initialization.

**Algorithm 1** The mirrored LeCun's initialization

---
1: Initialize all the weights using LeCun's initialization: $w_{i,j}^0 \sim N(0, \frac{1}{d})$, $v_i^0 \sim N(0, \frac{1}{m})$, for $i = 1, \ldots, m/2$, $j = 1, \ldots, d$.
2: Set $(w_{\frac{m}{2}+1}^0, \ldots, w_m^0) \leftarrow (w_1^0, \ldots, w_{\frac{m}{2}}^0)$, and set $(v_{\frac{m}{2}+1}^0, \ldots, v_m^0) \leftarrow (-v_1^0, \ldots, -v_{\frac{m}{2}}^0)$

---

Throughout the paper, we will make the following assumptions.

**Assumption 1.** *For $f(x; \theta)$ in (1), we assume its width $m$ is an even number, and $m \geq \frac{2n}{d}$.*

**Assumption 2.** *We assume the activation function $\sigma(\cdot) : \mathbb{R} \to \mathbb{R}$ is analytic and L-lipschitz continuous, its zero set only contains 0: $\{z | \sigma(z) = 0\} = \{0\}$. In addition, there are infinitely many non-zero coefficients in the Taylor expansion of $\sigma(\cdot)$.*

**Assumption 3.** *$x_1, \cdots, x_n$ are independently sampled from a continuous distribution in $\mathbb{R}^d$.*

When $d > 2$, Assumption 1 can be applied to narrow networks [5] with $m < n$. Assumption 2 covers many commonly used activation functions such as sigmoid, softplus, and Tanh, but it does not cover ReLU since it is nonsmooth.

### 4.2 Expressivity Analysis

In this section, we prove that narrow neural networks (which are still over-parameterized) have strong expressivity. Further, the zero-training-error solution is surrounded by a good landscape with no local-min or saddles, which motivates our trainability analysis in the following sections.

**Theorem 1.** *Suppose Assumption 1, 2, and 3 holds. If the neural network $f(x; \theta) = \sum_{j=1}^m v_j \sigma(w_j^T x)$ is initialized at the mirrored LeCun's initialization given in Algorithm 1, with $\theta^0 = (w^0, v^0)$, then there exists $\epsilon_0 > 0$ such that for any $\epsilon \leq \epsilon_0$, there exists a $w \in B_\epsilon(w^0) = \{w \mid \|w - w^0\|_F \leq \epsilon\}$ and a entry-wise non-zero $v$, such that with probabilty 1 of choosing $\{(x_i, y_i)\}_{i=1}^n$ and $\theta_0$, the output of $f$ will be exactly the groundtruth label:*

$$f(x_i; \theta) = \sum_{j=1}^m v_j \sigma(w_j^T x_i) = y_i, i = 1, \cdots, n. \tag{5}$$

*In addition, every stationary point $\theta^* = (w^*, v^*)$ (i.e., the gradient is zero) is a global-min of (2) with zero loss if it satisfies $w^* \in B_\epsilon(w^0)$ and $v^*$ is entry-wise non-zero.*

**Remark 1.** Theorem 1 emphasizes the role of hidden weights of a neural network: it is a key ingredient for strong expressivity. When $m < n$, if we fix all the $w = w^0$, the range space of the feature matrix $\Phi(w^0)$ does not cover the whole $\mathbb{R}^n$ space, so there always exists a label $y$, such that no $v^*$ can be found that perfectly maps the input to $y$. However, a small tolerance of the movement of $w$ will let $f(x; \theta)$ perfectly fit any input-label pair, so the movement of $w$ is vitally important. The free perturbation of $w$ serves as an effective remedy against the limited expressivity.

**Remark 2.** We emphasize that Theorem 1 holds for "any small enough $\epsilon$" instead of "any $\epsilon$". Therefore, Theorem 1 only states "there is no spurious local-min *locally*. It is still possible that on the *global* landscape results there "exists bad local-min" (e.g. Ding et al. [14]).

We comment a bit more on the maximum required size of $\epsilon$. In our proof in Appendix E.1, it should not exceed the the radius of the region where the Jacobian $J(w; v)$ stays full-rank (the yellow-shaded area in Figure 1). To briefly summarize, the maximum radius is (linearly) proportional to the minimum singular value of the initial Jacobian $J(w^0; v^0)$. Technical details on the size of this radius can be seen in [18, Remark 4.1].

*Proof sketch.* Theorem 1 consists of two arguments: (i) there exists a global-min with zero loss, (ii) in the neighborhood of initialization, every stationary point is a global-min. A detailed proof is relegated to Appendix E. We outline the main idea below.

---
[5]When $d = 1, 2$, all our results still hold; nevertheless, the required width $m \geq n, 2n$, thus it does not belong to the "narrow" setting we defined earlier in Section 2 (which requires $m \in [n/d, n)$).

To argue (i), the key idea is to use the Inverse Function Theorem (IFT), which is stated in Appendix E.1. According to IFT, as long as an $n \times n$ submatrix of $J(w^0; v^0)$ is invertible, then for any $y \in \mathbb{R}^n$ and any small enough $\epsilon$, there exists a $w^* \in B_\epsilon(w^0)$ whose prediction output $f(w^*; v^0) \propto y - f(w^0; v^0)$. Additionally, since $f(w^0; v^0) = 0$, we have $f(w^*; v^0) \propto y$. Once this is shown, then we just need to scale all the outer weight $v_j$ uniformly and the output will be exactly $y$ since $f(w^*; v)$ is linear in $v$.

To argue (ii), recall that all the stationary points $\theta^* = (w^*, v^*)$ satisfy $\nabla_w \ell(\theta^*) = J(w^*; v^*)^T(f(w^*; v^*) - y) = 0$. Therefore, if $J^T(w^*; v^*) \in \mathbb{R}^{md \times n}$ is of full column rank, the stationary point $\theta^* = (w^*, v^*)$ is a global minimizer with $\ell(\theta^*) = 0$. The desired full-rankness condition is true because: (i) $J(w^0; v^0)$ is full rank w.p.1 at initialization, (ii) $w^*$ will not leave the small neighborhood $B_\epsilon(w^0)$, so the dynamics of $w$ stays inside the manifold of full-rank Jacobian.

$\square$

The proof of Theorem 1 relies on the full-rankness of the Jacobian matrix. We note that such full-rankness holds for *both* the mirrored and the regular LeCun's initialization (the case for the regular one can be proved using the same technique). So why do we insist on shifting the initial output to 0? Simply put, the "randomness of weights" contributes to the "full-rankness", while the 'shifting' allows us to gain local representation power by applying IFT properly. To be more specific, $f(w^0, v^0) = 0 \in \mathbb{R}^n$ is important because it is surrounded by all possible directions pointed from $0 \in \mathbb{R}^n$, so for any $y \in \mathbb{R}^n$, IFT claims that there exists at least one $w^* \in B_\epsilon(w^0)$, s.t. $f(w^*; v^0) \propto y - f(w^0; v^0) = y$, therefore, $f(w^*; v^*)$ can perfectly match $y$ by scaling $v^0$ with some constant (Figure 3(a) illustrates this case when $n = 2$).

In contrast, if we use IFT around *regular* LeCun's initialization, the existence of $f(w^0; v^0)$ on the right hand side resists us from scaling $v^0$ like before (Figure 3(b) illustrates this case). As such, Theorem 1 does not hold for *any* small $\epsilon$ around the regular initialization. This is also revealed in our experiments in Section 5.1: training fails if we only search around the regular LeCun's initialization.

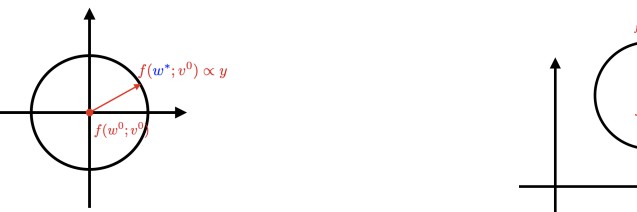

(a) The mirrored LeCun's initialization        (b) Regular LeCun's initialization

Figure 3: Examples of using Inverse Function Theorem (IFT) under different initialization strategies. The illustration here is for $n = 2$. For (a), scaling $v^0$ will directly lead to zero loss, while it is not true for (b) due to the non-zero $f(w^0; v^0)$.

The idea of zero initial output is also used in other recent works in Table 1. Despite the similar design, they use such an initialization for different purposes. In NTK regime, zero initial output helps eliminate the bias term and simplify the proof ([3, 11, 23, 24, 34] and [12]). Nguyen [49] also uses zero initial output, but their initialization is very different in that all the hidden layers will have high values while the last layer is assigned to 0. In this way, they manage to limit the movement of hidden layers without increasing the width. To our knowledge, this is the first time that the zero initial output has been linked to Inverse Function Theorem, by which the strong expressivity of a narrow neural network can be identified.

Table 1: Comparison of recent works considering zero initial output.

| Work | Width | Motivation |
|---|---|---|
| [3, 11, 23, 24, 34] | $m_{L-1} \to \infty$ | To avoid handling the bias term in the NTK regime |
| [12] | $m = \tilde{O}(n/d)$ | To avoid handling the bias term in the NTK regime |
| [49] | $m_{L-1} = O(n)$ | To ensure linear convergence via imbalanced weight |
| **Ours** | $m = O\left(\frac{n}{d}\right)$ | To achieve strong expressivity via Inverse Function Theorem |

### 4.3 Trainability Analysis

Despite the expressivity and good local properties stated in Theorem 1, in practical training, the weights can easily escape the nice neighborhood, especially when the width is not sufficiently large. To keep the hidden weights inside this nice region, an intuitive idea is to impose an explicit constraint, but it may suffer from bad local-min on the boundary with a very large loss [6]. This is supported by our experiments in Section 5, Figure 4, (b): when we add the hidden-weight constraint directly to the regular training regime (i.e., let $\|w - w^0\|_F \le \epsilon$), it fails to find a low-cost solution when $\epsilon$ and width become small.

In this sense, we need to design a constrained problem, such that all the KKT points will have small training loss, including those on the boundary. Fortunately, Theorem 1 suggests one such formulation. Recall in the proof of Theorem 1, we construct a zero-loss global-min by scaling $v^0$, so $v^*$ still follows the pairwise-opposite pattern. Inspired by this, we consider the following neural network (we abuse the notation of $f(x; \theta)$, $f(x; w, v)$ and $v = (v_1, \ldots, v_{\frac{m}{2}})$ here):

$$f(x; w, v) = \sum_{j=1}^{\frac{m}{2}} v_j \left( \sigma(w_j^T x) - \sigma(w_{j+\frac{m}{2}}^T x) \right). \tag{6}$$

Note that the optimization variable for the outer layer is only $v = (v_1, \ldots, v_{\frac{m}{2}})$, and the rest of the outer weights are automatically set to be $-v$. Despite the change of $v$, Theorem 1 still applies since it does not have any specific requirement on $v$. We then use (6) to formulate the following problem (7):

$$\min_{\theta} \ell(\theta) = \frac{1}{2} \sum_{i=1}^{n} (y_i - f(x_i; \theta))^2, \quad \text{s.t.} \qquad w \in B_\epsilon(w^0), \ v \in B_{\zeta, \kappa}(v) \tag{7}$$

where $f(x_i; \theta)$ is in the form of (6),

$B_\epsilon(w^0) := \{w \mid \|w - w^0\|_F \le \epsilon\}$,

$B_{\zeta, \kappa}(v) := \{v \mid v \ge \zeta \mathbf{1} \text{ and } v_j/v_{j'} \le \kappa, \forall (j, j') \in \{1, \cdots, m\}, \text{where } \zeta > 0, \ \kappa < \infty.\}$.

Here, $\zeta > 0$ is a small constant that keeps the entries of $v$ away from zero, which is an essential requirement of Theorem 1. The requirement of $v_j/v_{j'} \le \kappa < \infty$ allows all entries of $v$ to be uniformly large, but it rules out the case when some entries are much larger than others. Instead of regarding all these requirements of $B(v)$ as prior assumptions, we formulate them into the constraints in the problem, so all the iterates in the practical training algorithm will strictly follow these requirements. In Theorem 2, we show that every KKT point of problem (7) implies the near-global optimality for the unconstrained training problem (2).

**Theorem 2.** *Suppose Assumption 1, 2, and 3 hold and assume $\theta^0 = (w^0, v^0)$ as given in Algorithm 1. Then every KKT point $\theta^* = (w^*, v^*)$ of (7) is an approximate global-min w.p.1., that is:*

$$\ell(w^*, v^*) = O(\epsilon^2). \tag{8}$$

The proof of Theorem 2 is based on the special structure of neural network $f(x; \theta)$, including the linear dependence of $v$ and the mirrored pattern of parameters. To better illustrate our proof idea, we provide a user-friendly proof sketch in Appendix F.1. Detailed proof can be seen in Appendix F.2.

Theorem 2 motivates a training method to reach small loss. We highlight three new ingredients that is not used in regular neural net training: the mirrored initialization, the pairwise structure of $v$ in (7), and the constrained parameter movement. Combining these elements with Projected Gradient Descent (PGD), we propose a new training regime in Algorithm 2 in Appendix H.1. Thorough numerical results are provided in the following sections to demonstrate the efficacy of Algorithm 2.

### 4.4 Discussion: Extension to Deep Networks

In the previous sections, we analyze the trainability and expressivity of narrow 1-hidden-layer networks. We find it possible to extend the previous analysis to deep nets, and we already have some preliminary results. Due to space constraints, more relevant discussions are deferred to Appendix G.

## 5 Experiments

In this section, we provide empirical validation for our theory. Specifically, we compare the performance of two training regimes [6]:

---

[6]We call it "training regime" instead of "training method" since we use a different formulation as well a different algorithm compared to standard SGD.

**(1) Our training regime:** we optimize a constrained problem (7) by using PGD (projected gradient descent), starting from the mirrored LeCun's initialization. (See Algorithm 2 in Appendix H.1.)

**(2) Regular training regime:** optimize an unconstrained problem (2) by using GD-based methods, starting from LeCun's initialization.

**Main ingredients of our algorithm.** As shown in Theorem 2, all KKT points in our training regime have small empirical loss. To reach such KKT points, we use Projected Gradient Descent (PGD) (see Bertsekas [6]). Even though we do not provide convergence analysis, PGD can, empirically, converge to a KKT point with proper choice of stepsize. We outline the proposed training regime in Algorithm 2 in Appendix H.1. To briefly summarize, there are three key ingredients in Algorithm 2: the mirrored initialization; the pairwise structure of $v$ in (7); and the PGD algorithm. Each of these changes only involves a few lines of code changes based on the regular training. We demonstrate the `PyTorch` implementation of these changes in Appendix H.1.

**Better training and test error.** To evaluate our theory in terms of training error, we conduct experiments on synthetic dataset (shown in Section 5.1) and random-labeled CIFAR-10 [31] (shown in appendix H.6). We further observe the strong generalization power of Algorithm 2, even though it is not yet revealed in our theory. Our training regime brings higher or competitive test accuracy on (Restricted) ImageNet [58] (shown in Section 5.2), MNIST [33], CIFAR-10, CIFAR-100 [31] (shown in Appendix H). Detailed experimental setup are explained in Appendix H.2.

As a side note, for all the experiments in our training regime, we observe that $v$ never touches the boundary of $B_{\zeta,\kappa}(v)$ when $\kappa = 1, \zeta = 0.001$. So we can regard problem (7) as an unconstrained problem for $v$, and PGD only projects the hidden weights $w$ into $B_\epsilon(w^0)$. When $\epsilon = 1000$, problem (7) degenerates into an unconstrained problem (but still different from the regular training due to the changes in the structure of $v$ and initialization).

## 5.1 Training Error on The Synthetic Dataset

On the synthetic regression dataset, we train 1-hidden-layer networks under different widths and different training settings, the final training errors are shown in Figure 4. We explain as follows.

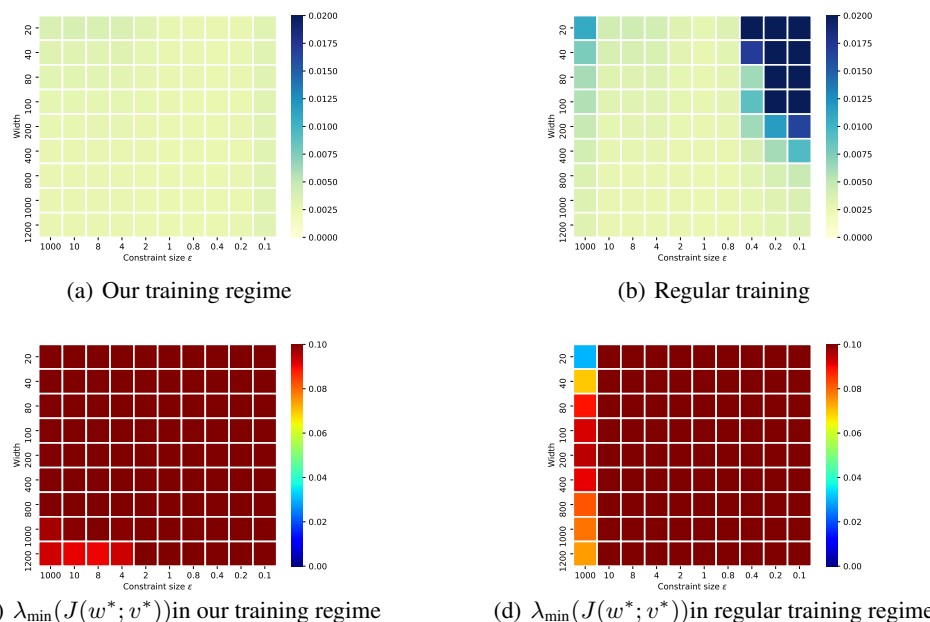

(a) Our training regime      (b) Regular training

(c) $\lambda_{\min}(J(w^*; v^*))$ in our training regime      (d) $\lambda_{\min}(J(w^*; v^*))$ in regular training regime

Figure 4: Synthetic data: training losses and $\lambda_{\min}(J(w^*; v^*))$ in different width & hidden-weight constraint size $\epsilon$.

As for our training regime, we try different hidden-weight constraint size $\epsilon$, these results can be seen in Figure 4, (a). Accordingly, the performance of the regular training regime can be seen in the 1st column in Figure 4, (b). As argued above, $\epsilon = 1000$ degenerates PGD into unconstrained GD, so this column shows the results for regular training regime. As for the rest of the columns in the Figure 4,

(b), we try to investigate an ablation study: "What will happen if we directly add constraint $B_\epsilon(w^0)$ on $w$, and use PGD without any modification of the initialization & network structure?" In each block in Figure 4, a grid search of step-size is performed to ensure the convergence of algorithms. The key messages from this set of experiments are summarized as follows.

First, our training regime performs well regardless of width, yet regular unconstrained training fails when the network is narrow (1st column in Figure 4 (b)).

Second, it does not work when we directly impose the hidden-weight constraint of (7) on the regular training regime (and PGD is used accordingly), as it fails to find a low-cost solution when $\epsilon$ and width become small. There are two possible causes: perhaps there is no global-min inside the ball, or it converges to bad local-min on the boundary. In contrast, our training regime always finds a low-cost solution with any choice of $\epsilon$. Therefore, we suggest *not* to directly add constraint and use PGD. Instead, when using PGD, it is better to utilize the mirrored initialization & the pairwise structure of $v$ in (7) (see Figure 4 (a)).

Furthermore, Figure 4, (c) & (d) depict $\lambda_{\min}(J(w^*; v^*))$, i.e. the minimum singular value of the Jacobian at the stationary (or KKT) points. As expected in Section 1, when the width is small, regular training regime (when $\epsilon = 1,000$) has trouble controlling the parameter movement, and it is likely to get trapped at a stationary point with a singular Jacobian matrix, leading to a large loss despite the convergence. However, this is not an issue in our training regime.

## 5.2  Test Accuracy on R-ImageNet

We check the test accuracy (not just the training accuracy) of the proposed method on R(Restricted)-ImageNet ([58]). R-ImageNet is a subset of ImageNet with resolution $224\times224$, and it is widely used in various papers (e.g. [68], [27]). Detailed description can be found in Appendix H.2. In both training regimes, experiments are conducted on ResNet-18 [22], which contains 4 CNN blocks and 1 fully connected output layer. To compare the performance under different widths, we shrink the number of channels in the final CNN block gradually from 512 to 64 (note that we will call "the number of channels" as the "width" in CNN, this is a straightforward extension of the width in FCN). In our training regime, we apply all the constraints in problem (7) to the final CNN block & the output layer.

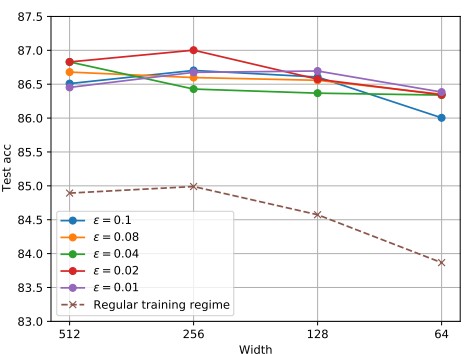

Figure 5: R-ImageNet: test accuracy under our training regime with different $\epsilon$ vs regular training regime. In x-axis, width stands for the number of channels in the final CNN block of ResNet-18. These results are averaged over 5 seeds.

There are two messages shown in Figure 5. First, our training regime outperforms regular training when using the standard ResNet-18 (i.e., width 512 in the final CNN block). Second, our training regime in the most narrow setting (width 64) performs quite close to the standard case (width 512). In comparison, regular SGD does not perform well in the narrow setting. More theoretical analysis on the generalization power will be considered as our future work.

## 6  Conclusion

In this work, we shed new light on both the expressivity and trainability of narrow networks. Despite the limited number of neurons, we prove that the network can memorize $n$ samples, and it can be provably trained to approximately zero loss in our training regime. We notice some interesting questions by reviewers and colleagues. We provide further discussion on these questions in Appendix A, they may be intriguing for general readers.

Finally, there are several important future directions. First, we empirically observe that our training regime brings strong generalization power, more theoretical analysis will be interesting. Second, our current analysis is still limited to 1-hidden-layer networks, we are trying to extend it to deep ones. Third, the algorithmic convergence analysis to reach the KKT point is imperative.

## Acknowledgments and Disclosure of Funding

We would like to thank Dawei Li for valuable and productive discussions. We want to thank the anonymous reviewers for their valuable suggestions and comments. We would also like to express our gratitude to Zeyu Qin, Jiancong Xiao and Congliang Chen for the support of R-ImageNet and CIFAR experiments. M. Hong is partially supported by an NSF grant CMMI-1727757, and an IBM Faculty Research Award. The work of Z.-Q. Luo is supported by the National Natural Science Foundation of China (No. 61731018) and the Guangdong Provincial Key Laboratory of Big Data Computation Theories and Methods.

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
