## Appendix

## Potential Negative Societal Impacts

In this paper, we discuss the expressivity and trainability of narrow neural networks. This paper provides a new understanding from the theoretical study, and such new insights will inspire a better training approach for neural networks with a small number of parameters. In industrial applications, several aspects of impact can be expected: training a huge neural network is at the expense of heavy computational burdens and large power consumption, which is a big challenge for embedded systems and small portable devices. In this sense, our work sheds new light on training narrow networks with much fewer parameters, so it will save energy for AI industries and companies. On the other hand, if everyone can afford to train powerful neural networks on their cell phone, then it will be a potential threat to most famous companies & institutes who are boast of their exclusive computational advantages. Additionally, there are chances that neural networks will be used for illegal usage.

## Appendix Organization

The Appendix is organized as follows.

- Appendix A provides some interesting questions by reviewers and colleagues. We provide further discussion on these questions, they may be intriguing for general readers.
- Appendix B provides more discussions on the literature.
- Appendix C provides more discussions on the related work Daniely [12].
- Appendix D introduces all the notations that will occur in the proof.
- Appendix E and F provide detailed proof for Theorem 1 and Theorem 2, respectively.
- Appendix G discusses extending the current analysis to deep neural networks.
- Appendix H introduces the following contents. (i) the formal statement of our training regime, i.e., Algorithm 2. (ii) The `Pytorch` implementation for Algorithm 2. (iii) Experimental details and settings for all the experiments appear in the paper. (iv) More experiments.

## A    Some More Discussions

In this section, we organize some frequently asked questions by reviewers and colleagues. Many of these questions may also be intriguing for general readers. We are thankful for the valuable discussion and we would like to share these questions. Here are our answers from the authors' perspective.

**Q1:** *The paper merely focuses on the optimization aspect, that is, minimizing the training loss, and ignores the more important problem (from ML perspective) of generalization. It would have been helpful if the authors include a discussion on the implications of these results on the generalization error as well.*

**A1:** We agree that generalization is a very important issue for deep learning (and still largely mysterious). Much of recent effort is spent on explaining why a huge number of parameters can still lead to a small generalization gap. Despite this interesting line of research, for narrow networks, the risk of overfitting is much smaller (due to traditional wisdom that fewer parameters lead to a smaller generalization gap), thus generalization of narrow networks is probably less mysterious than wide networks. According to our experiments on various real datasets ( in Section 5.2 & Appendix H), our training regime provides competitive or even better generalization performance than the regular training.

We think for the current stage, it may be more imperative to resolve expressiveness and optimization issues so as to improve practical performance. That being said, we agree that the theoretical study of the generalization gap is an interesting next step for research, and we will study it in future works.

**Q2:** *To which extent the result can be extended to non-smooth activations, such as ReLU or powers of ReLU?*

**A2:** We think it is possible, but definitely not easy. A main reason that we analyze smooth activation is that we need to prove the full-rankness of Jacobian $J(w; v)$ in Theorem 1. We believe that, for narrow nets, the full-rankness of Jacobian $J(w; v)$ with both smooth and non-smooth activation can be proved, but at the current stage, we lack suitable techniques for non-smooth ones (at least it is hard to extend the current technique to ReLU). We briefly summarize the technical difference below.

1. For ReLU, every entry of the NTK matrix $(J(w; v)J(w; v)^T)$ has a closed-form solution when the width $m = \infty$, and the corresponding NTK matrix is full rank under mild data assumption. This nice property allows us to utilize concentration inequalities to keep the full-rankness of $J(w; v)J(w; v)^T$ under finite (but large enough) width. This idea is used in Du et al, [16] for neural nets with width $m = \text{Poly}(n)$. However, this "concentration-based" approach is sensitive to width, it may be difficult to extend to narrow cases.

2. As for smooth activation, we utilize an important property of analytic function (shown in Lemma E.2). Lemma E.2 is based the intrinsic property of *any* analytic function, instead of a "infinite-width" argument. Therefore, it casts a higher possibility to use it for narrow nets. This property is also used in Li et al. [35] to prove the full-rankness of $J(w; v)$ for neural nets with width $m = O(n)$. Further, we successfully extend this result to width $m < n$ (with more sophisticated analysis).

In summary, analyzing ReLU requires new techniques. It is hard to extend the current analysis to narrow nets with general non-smooth activation.

**Q3:** *The "trainability" results only allow the small movement of hidden weights. This, in essence, is not desirable as it does not allow learning representations which is crucial in deep learning. Also, this is conceptually similar to the requirements in the NTK / lazy training regime, with the difference that those results offer precise convergence rates.*

**A3:** We believe our analysis is different from "lazy training", based on the following reasons.

i) We would like to point out that "small movement" does not imply our method is similar to "lazy training". Chizat et al.[11] described"lazy training" as the situation where "these two paths remain close until the algorithm is stopped". Here, the two paths correspond to the trajectory of training the original model and the linearized model respectively. "Training in a neighborhood of initialization" is neither a sufficient nor a necessary condition of "lazy training". For wide nets, "moving in a neighborhood" and "lazy training" are also co-existent, not causal. Logically speaking, for narrow nets, the two paths can be rather different while the training appears in a neighborhood.

ii) We then argue for narrow nets, linearized trajectory and neural net trajectory have to be quite different. Note that the linearized model of narrow nets cannot fit arbitrary data. In fact, when width $m < n$, the feature matrix does not span the whole $\mathbb{R}^n$ space if we fix first-layer hidden weights $w = w^0$. Thus our training trajectory has to be rather different from the linearized model trajectory, so as to fit data. In contrast, for wide networks, random fixed features suffice to represent data, so staying close to the linearized trajectory (or even coincide) can lead to zero training error.

iii) In our setting, the movement of first-layer hidden weights is *necessary* (no matter small or big). So "feature learning" is critical for our algorithm to achieve small training error. For wide net analysis, the movement of first-layer hidden weights is NOT necessary for zero training error, so there is no need for "feature learning".

In summary, narrow networks are out of the scope of lazy training analysis and have extra difficulties, making our analysis rather different from lazy training analysis (despite some similarities such as using the full-rankness of Jacobian).

For completeness, we further explain the main differences between our work and the previous papers on wide nets, most of the following opinions are also expressed in Section 3.

1. expressivity is rarely an issue for wide-net papers. A wide enough network (width $m > n$ ), even linear, can always fit $n$ arbitrary data samples (can be proved using simple linear algebra, which is shown in Section 3). However, when $m < n$, the expressivity is questionable. Fortunately, our Theorem 1 provides a clean positive answer.

2. When $m < n$, we are not clear whether the claim "GD converges to global-min in narrow nets" is true or not. In practice, GD easily got stuck at a large-loss stationary point in narrow-net training, but the wide nets are much easier to reach near-0 loss. So what we did

is NOT proving GD works in narrow nets, but designing an algorithm and showing it works (at least under certain conditions).

3. The empirical motivation is different. Existing NTK papers tried to "explain" why wide networks work well. We aim to "design" methods for training narrow networks, a topic of great interest for practitioners with limited computation resources and on-device AI.

# B   Additional Related Work

We provide discussions on other related work (in addition to those closely related ones mentioned in the main body).

**Memorization of small-width networks.** Yun et al. [73] studies how many neurons are required for a multi-layer ReLU network to memorize $n$ samples. In particular, they proved that if there are at least three layers, then the number of neurons per layer needed to memorize the data can be $O(\sqrt{n})$. They also show that if initialized near a global-min of the empirical loss, then SGD quickly finds a nearby point with much smaller loss. However, they did not mention how to find such an initialization strategy.

**Convergence analysis of $O(n)$-width networks.** Zhou et al. [78] studies the local convergence theory of mildly over-parameterized 1-hidden-layer networks. They show that, as long as the initial loss is low, GD converges to a zero-loss solution. They further propose an initialization strategy that provably returns a small loss under a mild assumption on the width. However, their proposed initialization may be costly to find since it requires solving an additional optimization problem.

**Convergence analysis of narrow networks.** A few recent works [10, 29, 51] showed that under a "$\gamma$-margin neural tangent separability" condition, GD converges to global minimizers for training 2-layer ReLU net with width [7] $\text{poly}(\log n, \gamma)$. For certain special data distributions where $\gamma = \text{poly}(\log n)$, their width is $\text{poly}(\log n, \log \frac{1}{\epsilon})$. Nevertheless, for general data distribution, their width can be larger than $n$.

**Global landscape analysis.** There are many works on the global landscape analysis; see, e.g., [15, 30, 32, 36–40, 44, 50, 52, 61, 62, 69, 76] and the surveys [64, 65]. For networks with width less than $n$, the positive result on the landscape either requires special activation like quadratic activation [62], or special data distribution like linearly separable or two-subspace data [38]. For certain non-quadratic activations, it was shown that sub-optimal strict local minima can exist for networks with width less than $n$ [36]. These results are different from ours in the scope, since they discuss *global landscape* of unconstrained problems, while we discuss local landscape.

# C   More Discussion on the Related Work: Daniely [12]

Daniely [12] studies the expressivity of 1-hidden-layer networks. They provide two results under different scenarios: to memorize $n(1 - \epsilon)$ random input-binary-label pairs via SGD, the required width is either

(i) $\tilde{O}\left(\frac{n}{\epsilon^2}\right)$ ([12, Theorem 5]); or

(ii) $\tilde{O}(n/d)$ ([12, Theorem 7]), where $\tilde{O}$ hides a factor that is dependent on $n$ and $d$.

In this section, we will explain our claim in Section 2 that their required width can be much larger than ours.

**Major differences**   In our work, our required width is smaller than $n$ (when $d > 2$). However, in either result (i) or (ii), their required width is often much larger than $n$. In fact, their width is actually at least $O\left(n^2\right)$ or larger for fixed $d$, if we consider the effect of $\epsilon$ (for their first bound) or the hidden factor in $\tilde{O}$ (for their second bound).

We will elaborate below.

---

[7]Their width also depends on desired accuracy $\epsilon$, and we skip the dependence here.

For their result (i), similar to Bubeck et al. [9], their required width grows with $\epsilon$. As the authors pointed out before their Theorem 7 on page 8, the required width is $\tilde{O}(n^3)$ if $\epsilon \leq 1/d$. This is why they add result (ii) in [12, Theorem 7].

For their result (ii), i.e., [12, Theorem 7] their required width is roughly $O(n/d\,(\log(d\log n))^{\log n})$. Compared to the desired bound $O(n/d)$, their actual bound contains an additional factor of roughly $(\log(d\log n))^{\log n}$. We would like to stress that the additional factor is *not* a constant factor or a log-factor, but more like a "super-polynomial-factor". As a result, the required width is actually at least $O(n^2/d)$ or larger. The detailed explanations are provided next, and briefly summarized below.

- In Appendix C.1, we will explain why there exists an extra "$(\log(d\log n))^{\log n}$" term in their required width bound. In [12, Theorem 7], their statement only mentioned $\tilde{O}(n/d)$ but not the exact expression.
- In Appendix C.2, we will explain why their bound is at least $O(n^2)$ or larger for fixed $d$.
- In Appendix C.3, we will summarize some other differences.

## C.1  Identifying a More Precise Width Bound

To find a more precise width bound of [12, Theorem 7], we tracked the proof of [12] as follows (note that their width $q$ is our $m$ and their sample size $m$ is our $n$, and we will use our notation below).

**The desired result.** Their goal is to memorize the $n(1-\epsilon)$ random input-label pairs via SGD.

**Notation.** They consider binary labels, i.e., $y_i = \{+1, -1\}$ for $i = 1, \cdots, n$. They consider a feature vector $\Psi_{\boldsymbol{\omega}}(\mathbf{x}_i) \in \mathbb{R}^{dm}$, where $\boldsymbol{\omega} \in \mathbb{R}^{d \times m}$. The predictor (classifier) is in the form of $f_{\boldsymbol{\omega}, \mathbf{v}}(\mathbf{x}) := \mathbf{v}^T \Psi_{\boldsymbol{\omega}}(\mathbf{x}_i)$, where $\mathbf{v} \in \mathbb{R}^{dm}$.

**Step 1 The desired result holds if there exists v such that a certain condition holds.**

More specifically, they have shown that the desired result holds if the following claim holds: there exists certain $\mathbf{v}$, such that w.h.p. over the dataset and $\boldsymbol{\omega} \in \mathbb{R}^{d \times m}$,

$$\langle \mathbf{v}, \Psi_{\boldsymbol{\omega}}(\mathbf{x}_i) \rangle = y_i + o(1), \quad \forall i = 1, \cdots, n. \tag{C.1}$$

Thus, the goal becomes to find $\mathbf{v}$ such that (C.1) holds. This goal is stated in the first paragraph of Section 4.3, page 15. Though they did not define it explicitly, $o(1)$ in (C.1) means " a constant smaller than 1".

The relation between (C.1) and the desired result is independent of the calculation of the width bound. Anyhow, for completeness, we briefly explain why (C.1) leads to the desired result (most readers can skip the paragraph). First, (C.1) implies that $f$ memorized the dataset for this $(\mathbf{v}, \Psi_{\boldsymbol{\omega}})$. In fact, if (C.1) holds, then $f_{\boldsymbol{\omega}, \mathbf{v}}(\mathbf{x}_i)$ has the same sign as $y_i + o(1)$. Note that $y_i + o(1)$ shares the same sign as $y_i$ because $y_i$ is assumed to be either $+1$ or $-1$ and $o(1)$ is a constant smaller than 1. Together we conclude that $f_{\boldsymbol{\omega}, \mathbf{v}}(\mathbf{x}_i)$ shares the same sign as $y_i$ for any $i$, which means the predictions $\mathrm{sign}(f_{\boldsymbol{\omega}, \mathbf{v}}(\mathbf{x}_i)) = y_i$. Second, such a $\mathbf{v}$ can be reached approximately by SGD (see Algorithm 2 on page 10 of [12]). This requires an argument that we skip here.

**Step 2: There exists v which satisfies a relation specified below.**

More specifically, they proved the following result.

**Theorem C.1** (Theorem 16, page 16, Daniely [12])**.** *W.p.* $1 - \delta - 2^{-\Omega(d)}$ *over the choice of dataset and* $\boldsymbol{\omega}$, *there exists a* $\mathbf{v}$ *such that the following for all* $i$:

$$\langle \mathbf{v}, \Psi_{\boldsymbol{\omega}}(\mathbf{x}_i) \rangle = f_{\boldsymbol{\omega}}(\mathbf{x}_i) = y_i + O\left(\frac{(\log(d/\delta))^{\frac{c'}{2}}}{d}\right) + O\left(\sqrt{\frac{d^{c-1}(\log(d/\delta))^{c'+2}}{m}}\right). \tag{C.2}$$

Note that in Theorem C.1, they did not explicitly write out the expression of $c$ and $c'$. The definition of these two constants can be seen in Section 3.2 and Section 4.3 of C.1 respectively. We will discuss them in Step 3.

**Step 3**: **Identifying a condition on** $m$ **so that the obtained bound** (C.2) **in Step 2 implies the desired bound** (C.1) **in Step 1.**

We demonstrate the detailed derivation next.

Step 3.0: To achieve the goal of (C.1), the 3rd term of (C.2) needs to be no more than 1. This term is the crucial part to derive the width bound on $m$. We provide detailed analysis as follows.

Step 3.1: Now we need to make sure the 3rd term of (C.2) is no more than 1, which is equivalent to (ignoring constant-factors)

$$m \geq d^{c-1}(\log(d/\delta))^{(c'+2)}, \tag{C.3}$$

where $c' \geq 4c + 2$. This definition of $c'$ can be seen in the first paragraph of Section 4.3 on page 15; $c, \delta$ are specified in the next two steps.

Step 3.2: Identify $\delta = 1/\log n$. This is specified in the paragraph below Theorem 16, page 16. Plugging it into (C.3), we have

$$m \geq d^{c-1}(\log(d \log n))^{(c'+2)}. \tag{C.4}$$

Step 3.3: Identify $c = \log n / \log d$. The definition of $c$ first appeared in the first paragraph of Section 3.2, where they assume the number of samples is $n = d^c$. This definition is used in the first paragraph of Section 4.3, with a slightly different form $n/d = d^{c-1}$. This definition implies $c = \log n / \log d$. Further, we have

$$c' \geq 4c + 2 \geq 4c + 2 = 4 \log n / \log d + 2 \tag{C.5}$$

Plugging (C.5) and $d^{c-1} = n/d$ into (C.4), we have:

$$m \geq \frac{n}{d}(\log(d \log n))^{4 \log n / \log d + 2}. \tag{C.6}$$

Finally, ignoring the numerical constants, the bound becomes

$$m \geq O(\frac{n}{d}(\log(d \log n))^{\log n / \log d}). \tag{C.7}$$

## C.2 Why The Bound is "Super-Polynomial"

Now we explain why their bound (C.7) is at least $O(n^2)$ or larger for fixed $d$. The bound (C.7) is a bit complicated as it depends on both $d$ and $n$. We are more interested in its dependence on $n$, thus we fix $d$ and analyze how it scales with $n$. With fixed $d$, the exponent $\log n / \log d$ can be simplified to $\log n$, thus the bound (C.7) can be simplified to

$$m \geq O(\frac{n}{d}(\log(d \log n))^{\log n}). \tag{C.8}$$

We will show that this bound (C.7) is at least $O(n^2)$ or larger for fixed $d$.

Define $B_1 = (\log d)^{\log n}$ and $B_2 = [\log(\log n)]^{\log n}$. From (C.8) we obtain

$$m \geq O(\frac{n}{d} \max\{B_1, B_2\}). \tag{C.9}$$

The extra factor $\max\{B_1, B_2\}$ is *not* a constant factor or a log-factor, but more like a "super-polynomial" factor on $n$, as explained below.

- For $B_1$, consider the fixed input dimension $d$ for two cases.
    - For $d$ satisfying $\log d > 2.7$ (i.e. $d > 15$), their required width is actually $O\left(\frac{n^2}{d}\right)$. This is an order of magnitude larger than our bound $O(n/d)$.
    - For $d$ satisfying $\log d > 2.7^2$ (i.e. $d > 1395$), the required width is actually $O\left(\frac{n^3}{d}\right)$, two orders of magnitude larger than $O(n/d)$.
- For $B_2$, when $n > 2.7^{2.7^{27}} \approx 2.2 \times 10^6$, we have $\left(\log(\log(n))^{\log n} > n\right.$. As a result, the required width is at least $O\left(\frac{n^2}{d}\right)$.

Theoretically speaking, it is not hard to prove that their required width can be larger than $n^k$ for any fixed integer $k$ (which is why we say their bound is "super-polynomial"). Empirically speaking, a calculation of their bound for a real dataset can reveal how large it is. On CIFAR-10 dataset [31], $n = 50000, d = 3072$. Plugging these numbers into (C.7) (ignore $O(\cdot)$) we obtain $m \geq 58290499136 >> n$. In comparison, our required width is only $m \geq 2n/d \approx 33$, which is much smaller than $n = 50000$.

In summary, rigorously speaking, their required width $\tilde{O}(n/d)$ is not $O(n/d)$, but can be larger than $O\left(\frac{n^2}{d}\right)$ or even $O\left(\frac{n^3}{d}\right)$.

### C.3 Other Differences

Besides the above discussion, there are some other differences between Daniely [12] and our work.

First, they analyze SGD, and we analyze a constrained optimization problem and projected SGD. This may be the reason why we can get a stronger bound on width. In the experiments in Section 5, we observe that SGD performs badly when the width is small (see the first left column in (b), Figure 4). Therefore, we suspect an algorithmic change is needed to train narrow nets with such width (due to the training difficulty), and we indeed propose a new method to train narrow nets.

Second, they consider binary $\{+1, -1\}$ dataset, while our results apply to arbitrary labels. In addition, their proof seems to be highly dependent on the fact that the labels are $\{+1, -1\}$, and seems hard to generalize to general labels.

## D  Definition and Notations

Before going through the proof details, we restate some of the important notations that will repeatedly appear in the proof, the following notations are also introduced in Section 4.1.

We denote $\{(x_i, y_i)\}_{i=1}^n \subset \mathbb{R}^d \times \mathbb{R}$ as the training samples, where $x_i \in \mathbb{R}^d$, $y_i \in \mathbb{R}$. For theoretical analysis, we focus on 1-hidden-layer neural networks $f(x; \theta) = \sum_{j=1}^m v_j \sigma\left(w_j^T x\right) \in \mathbb{R}$, where $\sigma(\cdot)$ is the activation function, $w_j \in \mathbb{R}^d$ and $v_j \in \mathbb{R}$ are the parameters to be trained. To learn such a neural network, we search for the optimal parameter $\theta = (w, v)$ by minimizing the empirical loss:

$$\min_\theta \ell(\theta) = \frac{1}{2} \sum_{i=1}^n \left(y_i - f\left(x_i; \theta\right)\right)^2.$$

Sometimes we also use $\ell(w; v)$ or $f(w; x, v)$ to emphasize the role of $w$.

We use the following shorthanded notations:

- $x := (x_1^T; \ldots; x_n^T) \in \mathbb{R}^{n \times d}$, $y := (y_1, \ldots, y_n)^T \in \mathbb{R}^n$;
- $w := (w_1, \ldots, w_m)_{j=1}^m \in \mathbb{R}^{d \times m}$, $v := (v_1, \ldots, v_m)_{j=1}^m \in \mathbb{R}^m$;
- $w_{a,b}$ indicates the $b$-th component of $w_a \in \mathbb{R}^d$;
- $\theta^0 = (w^0, v^0)$ indicates the initial parameters. Unless otherwise stated, it means the parameters at *the mirrored* LeCun's initialization given in Algorithm 1;
- $f(w; v) := (f(x_1; w, v), f(x_2; w, v), \ldots, f(x_n; w, v))^T \in \mathbb{R}^n$, indicating the neural network output on the whole dataset $x$.
- Define $\ell \circ f = \frac{1}{2}\|y - f\|_2^2$.

We denote the Jacobian matrix of $f(w; v)$ w.r.t $w$ as

$$J(w; v) := \begin{bmatrix} \nabla_w f\left(w; x_1, v\right)^T \\ \vdots \\ \nabla_w f\left(w; x_n, v\right)^T \end{bmatrix} = \begin{bmatrix} v_1 \sigma'\left(w_1^T x_1\right) x_1^T & \cdots & v_m \sigma'\left(w_m^T x_1\right) x_1^T \\ & \vdots & \\ v_1 \sigma'\left(w_1^T x_n\right) x_n^T & \cdots & v_m \sigma'\left(w_m^T x_n\right) x_n^T \end{bmatrix} \in \mathbb{R}^{n \times md}.$$

We define the feature matrix

$$\Phi(w) := \begin{bmatrix} \sigma\left(w_1^T x_1\right), \ldots, \sigma\left(w_m^T x_1\right) \\ \vdots \\ \sigma\left(w_1^T x_n\right), \ldots, \sigma\left(w_m^T x_n\right) \end{bmatrix} \in \mathbb{R}^{n \times m}.$$

## E  Proof of Theorem 1

The proof of Theorem 1 consists of two parts: when the hidden weights $w$ is in the neighborhood of the mirrored LeCun's initialization, we have (i) There exists a global-min with 0 loss, (ii) every

stationary point is a global-min. We prove these two arguments respectively. The first part (argument (i)) can be seen in Appendix E.1, the second part (argument (ii)) can be seen in Appendix E.2.

### E.1 Proof of The First Part of Theorem 1

To prove the first part of Theorem 1, we use the Inverse Function Theorem (IFT) [18] at the mirrored LeCun's initialization. IFT is stated below.

**Theorem E.1** (Inverse function theorem (IFT)). *Let $\psi : U \to \mathbb{R}^n$ be a $C^1$ -map where $U$ is open in $\mathbb{R}^n$ and $w \in U$. Suppose that the Jacobian $J(w)$ is invertible. There exist open sets $W$ and $F$ containing $w$ and $\psi(w)$ respectively, such that the restriction of $\psi$ on $W$ is a bijection onto $F$ with a $C^1$ -inverse.*

Our overall proof idea is as follows: we will use IFT to show that: then for any $y \in \mathbb{R}^n$ and any small enough $\epsilon$, there exists a $w^* \in B_\epsilon(w^0)$ whose prediction output $f(w^*; v^0) \propto y - f(w^0; v^0)$. Additionally, since $f(w^0; v^0) = 0$, we have $f(w^*; v^0) \propto y$. Once this is shown, then we just need to scale all the outer weight $v_j$ uniformly and the output will be exactly $y$ since $f(w^*; v)$ is linear in $v$. More details can be seen as follows.

In our case, let $\psi = f(w; v^0)$ be the function of $w$, mapping from $\mathbb{R}^{md}$ to $\mathbb{R}^n$. It may appears that IFT cannot be directly applied since $md \geq 2n$ (cf. Assumption 1), so $f(w; v^0)$ is not dimension-preserved mapping. However, this issue can be alleviated by applying the IFT to *a subvector* of $w$, while fixing the rest of the variables.

More specifically, we denote $n = k_1 d + k_2$ with $k_1, k_2 \in \mathbb{N}$, and $w = (\tilde{w}^T, \tilde{w}'^T)^T$, where $\tilde{w} = (w_1^T, \cdots, w_{k_1}^T, w_{k_1+1,1}, \cdots, w_{k_1+1,k_2})^T \in \mathbb{R}^n$ and $\tilde{w}' = (w_{k_1+1,k_2+1}, \cdots, w_{k_1+1,d}, w_{k_1+2}^T, \cdots, w_m^T)^T \in \mathbb{R}^{md-n}$. Here, $w_{a,b}$ indicates the $b$-th component of $w_a \in \mathbb{R}^d$.

We now apply IFT to $f(\tilde{w}; v^0, \tilde{w}'^0) \in \mathbb{R}^n$ (this notation views $\tilde{w}$ as the variable and $v^0, \tilde{w}'^0$ are treated as parameters). Firstly, in Lemma E.1, we prove that w.p.1, the corresponding Jacobian matrix $J(\tilde{w}^0; v^0, \tilde{w}'^0) \in \mathbb{R}^{n \times n}$ is of full rank at the mirrored LeCun's initialization, so that the condition for IFT holds. Then, by IFT, there exist open sets $W$ and $F$ containing $\tilde{w}^0$ and $f(\tilde{w}; v^0, \tilde{w}'^0)$ respectively, such that the restriction of $f(\tilde{w}; v^0, \tilde{w}'^0)$ on $W$ is a bijection onto $F$. Here, we denote $\epsilon$ and $\delta$ as the radius of $W$ and $F$, respectively.

Now, since $f(\tilde{w}^0; v^0, \tilde{w}'^0) = f(w^0; v^0) = 0 \in \mathbb{R}^n$, set $F$ contains all possible directions pointed from the origin. That is to say, for any label vector $y \in \mathbb{R}^n$, we can always scale it using $\delta$, such that $\delta \frac{y}{\|y\|} \in F$, and then, by IFT , there exists a $\tilde{w}^* \in B_\epsilon(\tilde{w}^0) \subset W$ satisfying

$$f(\tilde{w}^*; v^0, \tilde{w}'^0) = \delta \frac{y}{\|y\|}. \tag{E.1}$$

Since $\tilde{w}$ is just the truncated version of $w$, (E.1) implies: there exists a $w^* \in B_\epsilon(w^0)$, s.t.

$$f(w^*; v^0) = \delta \frac{y}{\|y\|}. \tag{E.2}$$

Now, we scale the outer weight to $v^* = \frac{\|y\|}{\delta} v^0$ and the output will be exactly $y$, i.e.:

$$f(w^*; v^*) = y.$$

Therefore, the proof is concluded.

**Remark: "there exists an $\epsilon$" or "any small $\epsilon$"?** Readers may mention that IFT states "there exists a neighborhood with size $\epsilon$", however, Theorem 1 claims for "any small enough $\epsilon$". We would like to clarify that the statement of Theorem 1 is *not* a typo. Here is the reason: in the statement of IFT, "existence of a small neighborhood" will imply "IFT holds for any subset of this neighborhood", so actually, Theorem 1 holds for any small (enough) neighborhood with size $\epsilon$.

**Lemma E.1.** *Under Assumption 1, 2 and 3, as a function of $w$, $v$ and $x$, $J(\tilde{w}^0; v^0, \tilde{w}'^0) \in \mathbb{R}^{n \times n}$ is of full rank at the mirrored LeCun's initialization, w.p.1..*

*Proof of Lemma E.1.* Recall the Jacobian matrix of $f(\tilde{w}; v^0, \tilde{w}'^0)$ w.r.t. $\tilde{w}$:

$$J(\tilde{w}^0; v^0, \tilde{w}'^0) := \begin{bmatrix} \nabla_{\tilde{w}} f\left(\tilde{w}^0; x_1, v^0, \tilde{w}'^0\right)^T \\ \vdots \\ \nabla_{\tilde{w}} f\left(\tilde{w}^0; x_n, v^0, \tilde{w}'^0\right)^T \end{bmatrix} \in \mathbb{R}^{n \times n}.$$

We first consider a general case where $k_2 \neq 0$. Since $f(\tilde{w}; x, v, \tilde{w}') = f(w; x, v) = \sum_{j=1}^{m} v_j \sigma\left(w_j^T x\right)$, taking derivative w.r.t. $\tilde{w}$ yields $J(\tilde{w}^0; v^0, \tilde{w}'^0)$ equals to:

$$\begin{pmatrix} v_1^0 \sigma'\left(w_1^{0T} x_1\right) x_1^T & \cdots & v_{k_1}^0 \sigma'\left(w_{k_1}^{0T} x_1\right) x_1^T & v_{k_1+1}^0 \sigma'\left(w_{k_1+1}^{0T} x_1\right) x_{1,1} & \cdots & v_{k_1+1}^0 \sigma'\left(w_{k_1+1}^{0T} x_1\right) x_{1,k_2} \\ & & \vdots & & & \\ v_1^0 \sigma'\left(w_1^{0T} x_n\right) x_n^T & \cdots & v_{k_1}^0 \sigma'\left(w_{k_1}^{0T} x_n\right) x_n^T & v_{k_1+1}^0 \sigma'\left(w_{k_1+1}^{0T} x_n\right) x_{n,1} & \cdots & v_{k_1+1}^0 \sigma'\left(w_{k_1+1}^{0T} x_n\right) x_{n,k_2} \end{pmatrix}.$$
(E.3)

To prove the full-rankness of $J(\tilde{w}^0; v^0, \tilde{w}'^0)$, we need to show that w.p.1, $\det(J(\tilde{w}^0; v^0, \tilde{w}'^0)) \neq 0$. Here, $\det(J(\tilde{w}^0; v^0, \tilde{w}'^0))$ is an analytic function since the activation function $\sigma(\cdot)$ is analytic (see Assumption 2). Therefore, we borrow an important result of [47, Proposition 0] which states that the zero set of an analytic function is either the whole domain or zero-measure. The result is formally stated as the following lemma under our notation.

**Lemma E.2.** *Suppose that: as a function of $\tilde{w}$, $\tilde{w}'$, $v$ and $x$, $\det(J(\tilde{w}^0; v^0, \tilde{w}'^0)) : \mathbb{R}^{md+m+nd} \to \mathbb{R}$ is a real analytic function on $\mathbb{R}^{md+m+nd}$. If $\det(J(\tilde{w}^0; v^0, \tilde{w}'^0))$ is not identically zero, then its zero set $\Omega = \left\{\tilde{w}^0, v^0, \tilde{w}'^0, x \mid \det(J(\tilde{w}^0; v^0, \tilde{w}'^0)) = 0\right\}$ has zero measure.*

Based on Lemma E.2, in order to prove $\det(J(\tilde{w}^0; v^0, \tilde{w}'^0)) \neq 0$ w.p.1, we only need to prove it is not identically zero. To do so, we first transform $\det(J(\tilde{w}^0; v^0, \tilde{w}'^0))$ into its equivalent form:

$$(E.3) \quad = \quad \det\left([B_1, \cdots, B_{k_2}, C_{k_2+1}, \cdots, C_d]\right), \tag{E.4}$$

where

$$B_j = \begin{pmatrix} v_1^0 \sigma'\left(w_1^{0T} x_1\right) x_{1,j} & \cdots & v_{k_1+1}^0 \sigma'\left(w_{k_1+1}^{0T} x_1\right) x_{1,j} \\ & \vdots & \\ v_1^0 \sigma'\left(w_1^{0T} x_n\right) x_{n,j} & \cdots & v_{k_1+1}^0 \sigma'\left(w_{k_1+1}^{0T} x_n\right) x_{n,j} \end{pmatrix} \in \mathbb{R}^{n \times (k_1+1)}, \quad j = 1, \cdots, k_2,$$

$$C_j = \begin{pmatrix} v_1^0 \sigma'\left(w_1^{0T} x_1\right) x_{1,j} & \cdots & v_{k_1}^0 \sigma'\left(w_{k_1}^{0T} x_1\right) x_{1,j} \\ & \vdots & \\ v_1^0 \sigma'\left(w_1^{0T} x_n\right) x_{n,j} & \cdots & v_{k_1}^0 \sigma'\left(w_{k_1}^{0T} x_n\right) x_{n,j} \end{pmatrix} \in \mathbb{R}^{n \times k_1}, \quad j = k_2+1, \cdots, d.$$

In addition, $B_j$ can be further rewritten as:

$$B_j = \begin{pmatrix} B_{1,j} \\ \vdots \\ B_{k_2,j} \\ D_j \end{pmatrix} \in \mathbb{R}^{n \times (k_1+1)},$$

where for $i = 1, \cdots k_2$:

$$B_{i,j} = \begin{pmatrix} v_1^0 \sigma'\left(w_1^{0T} x_{(k_1+1)(i-1)+1}\right) x_{(k_1+1)(i-1)+1,j} & \cdots & v_{k_1+1}^0 \sigma'\left(w_{k_1+1}^{0T} x_{(k_1+1)(i-1)+1}\right) x_{(k_1+1)(i-1)+1,j} \\ & \vdots & \\ v_1^0 \sigma'\left(w_1^{0T} x_{(k_1+1)i}\right) x_{(k_1+1)i,j} & \cdots & v_{k_1+1}^0 \sigma'\left(w_{k_1+1}^{0T} x_{(k_1+1)i}\right) x_{(k_1+1)i,j} \end{pmatrix}$$

$$\in \mathbb{R}^{(k_1+1) \times (k_1+1)},$$

and

$$D_j = \begin{pmatrix} v_1^0 \sigma'\left(w_1^{0T} x_{(k_1+1)k_2+1}\right) x_{(k_1+1)k_2+1,j} & \cdots & v_{k_1+1}^0 \sigma'\left(w_{k_1+1}^{0T} x_{(k_1+1)k_2+1}\right) x_{(k_1+1)k_2+1,j} \\ & \vdots & \\ v_1^0 \sigma'\left(w_1^{0T} x_n\right) x_{n,j} & \cdots & v_{k_1+1}^0 \sigma'\left(w_{k_1+1}^{0T} x_n\right) x_{n,j} \end{pmatrix}$$

$$\in \mathbb{R}^{(n-(k_1+1)k_2) \times (k_1+1)}.$$

Similarly, $C_j$ can be further rewritten as:

$$C_j = \begin{pmatrix} C_{1,j} \\ \vdots \\ C_{k_2,j} \\ E_j \end{pmatrix} \in \mathbb{R}^{n \times (k_1)},$$

where for $i = 1, \cdots k_2$:

$$C_{i,j} = \begin{pmatrix} v_1^0 \sigma' \left( w_1^{0T} x_{(k_1+1)(i-1)+1} \right) x_{(k_1+1)(i-1)+1,j} & \cdots & v_{k_1}^0 \sigma' \left( w_{k_1}^{0T} x_{(k_1+1)(i-1)+1} \right) x_{(k_1+1)(i-1)+1,j} \\ \vdots & & \\ v_1^0 \sigma' \left( w_1^{0T} x_{(k_1+1)i} \right) x_{(k_1+1)i,j} & \cdots & v_{k_1}^0 \sigma' \left( w_{k_1}^{0T} x_{(k_1+1)i} \right) x_{(k_1+1)i,j} \end{pmatrix}$$

$$\in \quad \mathbb{R}^{(k_1+1) \times k_1}$$

and

$$E_j = \begin{pmatrix} v_1^0 \sigma' \left( w_1^{0T} x_{(k_1+1)k_2+1} \right) x_{(k_1+1)k_2+1,j} & \cdots & v_{k_1}^0 \sigma' \left( w_{k_1}^{0T} x_{(k_1+1)k_2+1} \right) x_{(k_1+1)k_2+1,j} \\ \vdots & & \\ v_1^0 \sigma' \left( w_1^{0T} x_n \right) x_{n,j} & \cdots & v_{k_1}^0 \sigma' \left( w_{k_1}^{0T} x_n \right) x_{n,j} \end{pmatrix}$$

$$\in \quad \mathbb{R}^{(n-(k_1+1)k_2) \times k_1}.$$

Therefore, we can rewrite $\det(J(\tilde{w}^0; v^0, \tilde{w}'^0))$ as the following form:

$$\det(J(\tilde{w}^0; v^0, \tilde{w}'^0)) = \begin{pmatrix} B_{1,1} & \cdots & B_{1,k_2} & C_{1,k_2+1} & \cdots & C_{1,d} \\ & & \vdots & & & \\ B_{k_2,1} & \cdots & B_{k_2,k_2} & C_{k_2,k_2+1} & \cdots & C_{k_2,d} \\ D_1 & \cdots & D_{k_2} & E_{k_2+1} & \cdots & E_d \end{pmatrix}. \tag{E.5}$$

Based on Lemma E.2, in order to prove $\det(J(\tilde{w}^0; v^0, \tilde{w}'^0)) \neq 0$ w.p.1., we only need to prove that, as a function of $\tilde{w}$, $\tilde{w}'$, $v$ and $x$, $\det(J(\tilde{w}^0; v^0, \tilde{w}'^0))$ is not identically zero. To do so, we just need to construct a dataset $x$ such that $(E.5) \neq 0$. We construct such $x := (x_1, \ldots, x_n) \in \mathbb{R}^{n \times d}$ in the following way:

(1) For $i = 1, \cdots, (k_1+1)$: $x_{i,j} = \begin{cases} \delta_i, & j = 1 \\ 0, & \text{otherwise} \end{cases}$ , where $\delta_i \neq \delta_{i'} \neq 0, \forall i, i'$.

(2) For $i = (k_1+1) + 1, \cdots, 2(k_1+1)$: $x_{i,j} = \begin{cases} \delta_i, & j = 2 \\ 0, & \text{otherwise} \end{cases}$ , where $\delta_i \neq \delta_{i'} \neq 0$, $\forall i, i'$.

(3) $\cdots$

(4) For $i = (k_2-1)(k_1+1) + 1, \cdots, k_2(k_1+1)$: $x_{i,j} = \begin{cases} \delta_i, & j = k_2 \\ 0, & \text{otherwise} \end{cases}$ , where $\delta_i \neq \delta_{i'} \neq 0, \forall i, i'$.

(5) For $i = k_2(k_1+1) + 1, \cdots, n$: $x_{i,j} = \begin{cases} 1, & j = i - k_1 k_2 \\ 0, & \text{otherwise} \end{cases}$ .

Under such a construction, (E.5) becomes the determinant of a block-diagonal matrix:

$$\det(J(\tilde{w}^0; v^0, \tilde{w}'^0)) = \det \begin{pmatrix} B_{1,1} & \cdots & 0 & 0 \\ & & \vdots & \\ 0 & \cdots & B_{k_2,k_2} & 0 \\ 0 & \cdots & 0 & E \end{pmatrix}, \tag{E.6}$$

where $E = [E_{k_2+1} \cdots E_d]$ is a square matrix in $\mathbb{R}^{(n-(k_1+1)k_2)\times(n-(k_1+1)k_2)}$. To prove $(E.6) \neq 0$, we need to prove $B_{1,1}, \cdots, B_{k_2,k_2}$ and $E$ are all full rank matrices.

As for the full-rankness of $E$, thanks to the construction (5), $E = [E_{k_2+1} \cdots E_d]$ now becomes:

$$
E_{k_2+1} = \begin{pmatrix} v_1^0 \sigma' \left( w_1^{0T} x_{(k_1+1)k_2+1} \right) & \cdots & v_{k_1}^0 \sigma' \left( w_{k_1}^{0T} x_{(k_1+1)k_2+1} \right) \\ 0 & \cdots & 0 \\ & \vdots & \\ 0 & \cdots & 0 \end{pmatrix} \in \mathbb{R}^{(n-(k_1+1)k_2)\times k_1},
$$

$$
E_{k_2+2} = \begin{pmatrix} 0 & \cdots & 0 \\ v_1^0 \sigma' \left( w_1^{0T} x_{(k_1+1)k_2+2} \right) & \cdots & v_{k_1}^0 \sigma' \left( w_{k_1}^{0T} x_{(k_1+1)k_2+2} \right) \\ & \vdots & \\ 0 & \cdots & 0 \end{pmatrix} \in \mathbb{R}^{(n-(k_1+1)k_2)\times k_1},
$$

and so on so for:

$$
E_d = \begin{pmatrix} 0 & \cdots & 0 \\ & \vdots & \\ 0 & \cdots & 0 \\ v_1^0 \sigma' \left( w_1^{0T} x_n \right) & \cdots & v_{k_1}^0 \sigma' \left( w_{k_1}^{0T} x_n \right) \end{pmatrix} \in \mathbb{R}^{(n-(k_1+1)k_2)\times k_1}.
$$

Since $md \geq 2n$ and $J(\tilde{w}^0; v^0, \tilde{w}'^0) \in \mathbb{R}^{n\times n}$ is the jacobian w.r.t. the first $n$ components of $w \in \mathbb{R}^{md}$, it only involves $w_1, w_2, \cdots w_{k_1+1}$ and it will not reach beyond $w_{\frac{m}{2}} \in \mathbb{R}^d$. Recall in the mirrored LeCun's initialization, we only copy the 2nd half of $w$: $(w_{\frac{m}{2}+1}^0, \ldots, w_m^0) \leftarrow (-w_1^0, \ldots, -w_{\frac{m}{2}}^0)$, that is to say, $w_1, w_2, \cdots w_{k_1+1}$ are independent Gaussian random variables, unaffected by the copying phase, similarly for $v_1, \cdots, v_{k_1+1}$.

In short, since Gaussian random variables take value 0 on a zero probability measure, and $\sigma(z) = 0$ only happens when $z = 0$ (see Assumption 2), we have $E = [E_{k_2+1} \cdots E_d]$ is full rank w.p.1.

As for the full-rankness of $B_{kk}$, $k = 1, \cdots, k_2$, we only need to prove $B_{11}$ is invertible, the proof of the rest of $B_{kk}$ are the same.

Under construction (1), we have:

$$
B_{1,1} = \begin{pmatrix} v_1^0 \sigma' \left( w_1^{0T} x_1 \right) \delta_1 & \cdots & v_{k_1+1}^0 \sigma' \left( w_{k_1+1}^{0T} x_1 \right) \delta_1 \\ & \vdots & \\ v_1^0 \sigma' \left( w_1^{0T} x_{(k_1+1)} \right) \delta_{k_1+1} & \cdots & v_{k_1+1}^0 \sigma' \left( w_{k_1+1}^{0T} x_{(k_1+1)} \right) \delta_{k_1+1} \end{pmatrix} \in \mathbb{R}^{(k_1+1)\times(k_1+1)}.
$$

Again, since Gaussian random variables take value 0 on a zero probability measure, and $\delta_i \neq 0$ for $i = 1, \cdots, k_1 + 1$, we only need to prove the following $\tilde{B}_{1,1}$ is full rank:

$$
\tilde{B}_{1,1} = \begin{pmatrix} \sigma' \left( w_1^{0T} x_1 \right) & \cdots & \sigma' \left( w_{k_1+1}^{0T} x_1 \right) \\ & \vdots & \\ \sigma' \left( w_1^{0T} x_{(k_1+1)} \right) & \cdots & \sigma' \left( w_{k_1+1}^{0T} x_{(k_1+1)} \right) \end{pmatrix} \in \mathbb{R}^{(k_1+1)\times(k_1+1)}.
$$

Next, we borrow the following lemma from [35, Proposition 1], which is the restatement under our notation (their original statement applies for deep neural network, here, we restate it for 1-hidden-layer case in Lemma E.3).

**Lemma E.3.** *Under Assumption 2, given an 1-hidden-layer neural network with width $m \geq n$, Let $\Omega = \{w \mid \mathrm{rank}\,(\Phi(w)) < \min\{m,n\}\}$, where $\Phi(w)$ is the hidden feature matrix*

$$
\Phi(w) := \begin{bmatrix} \sigma \left( w_1^T x_1 \right), \ldots, \sigma \left( w_m^T x_1 \right) \\ \vdots \\ \sigma \left( w_1^T x_n \right), \ldots, \sigma \left( w_m^T x_n \right) \end{bmatrix} \in \mathbb{R}^{n\times m}.
$$

*Suppose there exists a dimension $k$ such that $x_{i,k} \neq x_{i',k}, \forall i \neq i'$, then $\Omega$ is a zero-measure set.*

To prove the full-rankness of $\tilde{B}_{1,1}$, we regard it as the hidden feature matrix of an 1-hidden-layer neural network equipped with width $m' = k_1 + 1$ and activation function $\sigma'(z)$, which satisfies Assumption 2. In addition, recall $\delta_i \neq \delta_{i'} \neq 0$ for $\forall i, i'$, so $(x_1, \cdots, x_{k_1+1})$ satisfies the condition of Lemma E.3 w.p.1, and the sample size equals to the width $m' = k_1 + 1$. Therefore, all the assumptions are satisfied and Lemma E.3 directly shows that $\tilde{B}_{1,1}$ is invertible w.p.1..

Similarly, with the same proof technique, it can be shown that the rest of $B_{k,k}$ are also invertible w.p.1.. In conclusion, we have constructed a dataset $x$, such that (E.6) is non-zero w.p.1., which implies $\Omega = \left\{ \tilde{w}^0, v^0, \tilde{w}'^0, x \mid \det(J(\tilde{w}^0; v^0, \tilde{w}'^0)) = 0 \right\}$ has zero measure by Lemma E.2. In other words, under the joint distribution of $\tilde{w}^0$, $v^0$, $\tilde{w}'^0$ and $x$, $J(\tilde{w}^0; v^0, \tilde{w}'^0)$ is invertible w.p.1.. Recall $\tilde{w}^0$, $v^0$, and $\tilde{w}'^0$ all follow continuous distribution, furthermore, $x$ also follows a continuous distribution (Assumption 3), so $J(\tilde{w}^0; v^0, \tilde{w}'^0)$ is still invertible w.p.1. under the distribution of $x$, so the whole proof is completed.

When $n = k_1 d$, or equivalently, $k_2 = 0$, things become easier and we just need to change the size of $B_{kk}$ to $\mathbb{R}^{k_1 \times k_1}$, and there is no need to consider $C_{i,j}$, $D_j$ and $E_j$, the rest of the proof is the same, we omit it for brevity. $\qquad\square$

## E.2 Proof of The Second Part of Theorem 1

Suppose we have a 1-hidden-layer neural network $f(x; \theta) = \sum_{j=1}^{m} v_j \sigma\left(w_j^T x\right) \in \mathbb{R}$, and let $f(\theta) \in \mathbb{R}^n$ be output of $(x; \theta)$ on the dataset $x = (x_1, \cdots, x_n)$, let $J(w^*; v^*) \in \mathbb{R}^{n \times md}$ be its Jacobian matrix w.r.t. $w$ at the stationary point $\theta^* = (w^*, v^*)$, we have:

$$\nabla_w \ell(\theta^*) = J(w^*; v^*)^T (f(\theta^*) - y) = 0. \tag{E.7}$$

Therefore, as long as we can prove that $J(w^*; v^*)^T \in \mathbb{R}^{md \times n}$ is full column rank, the stationary point $\theta^*$ will become a global minimizer with $\ell(\theta^*) = 0$, so the proof is completed.

Now we prove the full-rankness of $J(w^*; v^*)$. Recall in Lemma E.1, we have proved that, as a function of $x$, $\det(J(\tilde{w}^0; v^0, \tilde{w}'^0)) \neq 0$ w.p.1., where $J(\tilde{w}^0; v^0, \tilde{w}'^0)$ equals to

$$\begin{pmatrix} v_1^0 \sigma'\left(w_1^{0T}x_1\right)x_1^T & \cdots & v_{k_1}^0 \sigma'\left(w_{k_1}^{0T}x_1\right)x_1^T & v_{k_1+1}^0 \sigma'\left(w_{k_1+1}^{0T}x_1\right)x_{1,1} & \cdots & v_{k_1+1}^0 \sigma'\left(w_{k_1+1}^{0T}x_1\right)x_{1,k_2} \\ & & \vdots & & & \\ v_1^0 \sigma'\left(w_1^{0T}x_n\right)x_n^T & \cdots & v_{k_1}^0 \sigma'\left(w_{k_1}^{0T}x_n\right)x_n^T & v_{k_1+1}^0 \sigma'\left(w_{k_1+1}^{0T}x_n\right)x_{n,1} & \cdots & v_{k_1+1}^0 \sigma'\left(w_{k_1+1}^{0T}x_n\right)x_{n,k_2} \end{pmatrix},$$
$$\tag{E.8}$$

Since $\|w^* - w^0\|_F \leq \epsilon$ and the determinant is a continuous function of $w$, we have $\det(J(\tilde{w}^*; v^0, \tilde{w}'^*)) \neq 0$ (w.p.1.) when $\epsilon$ is small. In addition, as we can see from (E.8), $(v_1^0, \cdots, v_{k_1+1}^0)$ are just constant terms in the corresponding columns, so the full-rankness of (E.8) still holds if we change $v_j^0$ to $v_j^* \neq 0$. In summary, $\det(J(\tilde{w}^*; v^*, \tilde{w}'^*)) \neq 0$ when $v^*$ is entry-wise non-zero, where $J(\tilde{w}^*; v^*, \tilde{w}'^*)$ equals to

$$\begin{pmatrix} v_1^* \sigma'\left(w_1^{*T}x_1\right)x_1^T & \cdots & v_{k_1}^* \sigma'\left(w_{k_1}^{*T}x_1\right)x_1^T & v_{k_1+1}^* \sigma'\left(w_{k_1+1}^{*T}x_1\right)x_{1,1} & \cdots & v_{k_1+1}^* \sigma'\left(w_{k_1+1}^{*T}x_1\right)x_{1,k_2} \\ & & \vdots & & & \\ v_1^* \sigma'\left(w_1^{*T}x_n\right)x_n^T & \cdots & v_{k_1}^* \sigma'\left(w_{k_1}^{*T}x_n\right)x_n^T & v_{k_1+1}^* \sigma'\left(w_{k_1+1}^{*T}x_n\right)x_{n,1} & \cdots & v_{k_1+1}^* \sigma'\left(w_{k_1+1}^{*T}x_n\right)x_{n,k_2} \end{pmatrix},$$
$$\tag{E.9}$$

Furthermore, $J(\tilde{w}^*; v^*, \tilde{w}'^*) \in \mathbb{R}^{n \times n}$ is nothing but a $n \times n$ submatrix of $J(w^*; v^*) \in \mathbb{R}^{n \times md}$. Now that $md \geq 2n > n$, $\det(J(\tilde{w}^*; v^*, \tilde{w}'^*)) \neq 0$ implies the full-row-rankness of $J(w^*; v^*)$ (w.p.1.). Thus the whole proof is completed.

# F Proof of Theorem 2

In this section, we provide both proof sketch and the detailed proof of Theorem 2. They can be seen in Appendix F.1 and Appendix F.2, respectively. For general readers, reading proof sketch in Appendix F.1 will help grasp our main idea.

### F.1 Proof Sketch of Theorem 2

*Proof sketch.* The proof is built on the special structure of neural network $f(x; \theta)$, including the linear dependence of $v$ and the mirrored pattern of parameters. Here, we describe our high level idea and the analysis roadmap. The proof consists of proving the following claims:

(I) every KKT point $\theta^*$ satisfies $\|\nabla_w \ell(w^*; v^*)\| = O(\epsilon)$;

(II) the gradient of $w$ always dominates the error term, i.e. $\|\nabla_w \ell(w; v)\|_2^2 = \Omega\left(\ell(w^*; v^*)\right)$, so we have $\ell(w^*; v^*) = O(\epsilon^2)$.

To prove claim (I), we only need to consider the case where $w^*$ is on the boundary and $\|\nabla_w \ell(w^*; v^*)\|_2 \neq 0$ (otherwise Theorem 2 automatically holds based on Theorem 1). In this case, by the optimality condition, taking $\eta \in \mathbb{R}$ as a small step size, we have

$$-\eta \nabla_w \ell(w^*; v^*) = \eta \frac{\|\nabla_w \ell(w^*; v^*)\|}{\|w^* - w^0\|}(w^* - w^0) = \tilde{\eta}(w^* - w^0), \tag{F.1}$$

where $\tilde{\eta} = \eta \frac{\|\nabla_w \ell(w^*; v^*)\|}{\|w^* - w^0\|}$. Now, our key observation is that, after moving along (F.1) from $w^*$, the change of the loss is not significant due to the special local structure of $f(w; v^*)$, therefore, $\|\nabla_w \ell(w^*; v^*)\|_2$ can be bounded. To be more specific, we denote $f^*$ as the neural network output at the KKT point $\theta^*$; denote $\bar{f}$ as the neural network output after taking a small step $\eta$ along the negative partial gradient direction of $w$; and denote $f'$ as an rough estimate of $\bar{f}$:

$$f^* \quad := \quad f(w^*; v^*) = J(w^0; v^*)w^* + R^*, \tag{F.2}$$

$$\bar{f} \quad := \quad f(w^* - \eta \nabla_w \ell(w^*; v^*); v^*) = (1 + \tilde{\eta})f^* - (1 + \tilde{\eta})R^* + \bar{R}, \tag{F.3}$$

$$f' \quad := \quad \bar{f}_{\text{lin}} + (1 + \tilde{\eta})R^* = (1 + \tilde{\eta})f^*, \tag{F.4}$$

where $\bar{f}_{\text{lin}}$ is the first-order Taylor approximation of $\bar{f}$, $R^*$ is the second-order Taylor residue of $f^*$ (similarly for $\bar{R}$). Additionally, (F.2), (F.3), and (F.4) are due to the the symmetric property of $w^0$, $v^0$ and $v^*$, so the bias terms in the Taylor expansion will vanish. Now, we compare the value of the loss on each of $f^*$, $\bar{f}$ and $f'$. Define $\ell \circ f = \frac{1}{2}\|y - f\|_2^2$, we prove the following crucial relationship:

$$\eta \|\nabla_w \ell(w^*; v^*)\|_2^2 \overset{(a)}{\leq} \ell \circ f^* - \ell \circ \bar{f} \overset{(b)}{\leq} \ell \circ f' - \ell \circ \bar{f} = \frac{1}{2}\left\|f' - \bar{f}\right\|_2 \left\|f' + \bar{f} - 2y\right\|_2 \overset{(c)}{=} O(\tilde{\eta}\epsilon^2) \tag{F.5}$$

Eq. (F.5) plays a key role in our analysis. Here, $(a)$ can be easily shown by applying Descent lemma in this local region, yet $(b)$ and $(c)$ are not that obvious. Recall in (F.2), (F.3), and (F.4), we know that: (i) although the location of $\bar{f}$ is unclear, $f'$ points at the same direction as $f^*$. (ii) As an estimator of $\bar{f}$, $f'$ is not far away from it, i.e. $\|\bar{f} - f'\|_2 = \|\bar{R} - (1 + \tilde{\eta})R^*\|_2$ only involves the second-order Taylor residue terms. With this observation, $(b)$ is proved in Lemma F.1 (stated below) by geometric properties, and $(c)$ is calculated in Lemma F.2 (stated below), so the relationship (F.5) is proved. Therefore, we have $\theta^*$ satisfies $\|\nabla_w \ell(w^*; v^*)\| = O(\epsilon)$ by plugging in $\eta = \frac{\epsilon \tilde{\eta}}{\|\nabla_w \ell(w^*; v^*)\|_2}$, and claim (I) is proved.

**Lemma F.1.** *Under the conditions of Theorem 2, we have $\ell \circ f' \geq \ell \circ f^*$, i.e., $\|f' - y\|_2^2 \geq \|f^* - y\|_2^2$.*

**Lemma F.2.** *Under the conditions of Theorem 2, we have: $\left\|f' - \bar{f}\right\|_2 \left\|f' + \bar{f} - 2y\right\|_2 = O(\epsilon^2)$.*

As for claim (II), it is true as long as $J(w; v)$ is of full row rank, which has been shown in Theorem 1. A more detailed proof of Lemma F.1 and F.2, as well as the proof of the whole Theorem 2 are in Appendix F.2.

*Proof sketch of Lemma F.1.* Here, we provide a proof sketch of Lemma F.1, we need to discuss the following cases:

(1) When $f^{*T}y \geq 0$: we prove by contradiction. Since $f^*$ is linear in $v^*$, $\|(1 + \tilde{\eta})f^* - y\|_2^2 < \|f^* - y\|_2^2$ implies $\|f(w^*; (1 + \tilde{\eta})v^*) - y\|_2^2 < \|f^* - y\|_2^2$, which means we can further reduce the loss by changing $v^*$ to $(1 + \tilde{\eta})v^*$, which is still feasible, we have a contradiction to the assumption that $(w,^* v^*)$ is a KKT point (see Figure 6, Middle).

(2) When $f^{*T}y < 0$: changing $v^*$ to $-v^*$ will further reduce the distance to $y$ (see Figure 6, Left), this is a contradiction to the fact that $v^*$ is a KKT point, so case (2) will not happen.

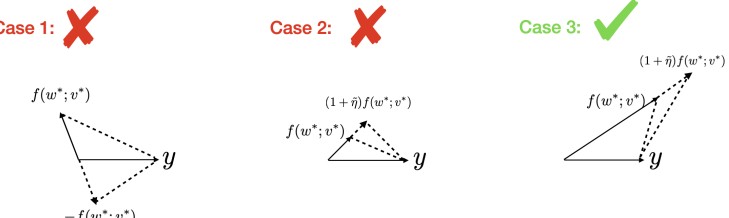

Figure 6: Geometrical illustration of three cases in Lemma F.1: comparing with $f^*$, $f' = (1 + \tilde{\eta})f^*$ will not further reduce the distance to $y$.

In conclusion, we always have $\ell \circ f' \geq \ell \circ f^*$ (see Figure 6, Right). $\qquad\square$

## F.2 Detailed Proof of Theorem 2

The proof of Theorem 2 consists of proving the following claims: under the setting of Theorem 2,

(I) every KKT point $\theta^*$ satisfies $\|\nabla_w \ell(w^*; v^*)\| = O(\epsilon)$.

(II) the gradient of $w$ always dominates the error term, i.e. $\|\nabla_w \ell(w; v)\|_2^2 = \Omega\left(l(w^*; v^*)\right)$, so we have $\ell(w^*; v^*) = O(\epsilon^2)$.

Note that the statement of claim (I) is not precise, there are chances that the constant terms will exponentially grow (will be discussed later). Nevertheless, it is just an intermediate result that helps provide a clearer big picture. our final result in claim (II) will be precise.

To prove claim (I), we only need to consider the case when $w^*$ is on the boundary of the constraint $B_\epsilon(w^0)$ (Theorem 2 automatically holds when $w^*$ is in the interior of $B_\epsilon(w^0)$).

Now, suppose $w^*$ is a non-zero-gradient KKT point on the boundary, by the optimality condition, its negative gradient direction should be along the same direction as $w^* - w^0$, therefore, if we further take a small step $\eta$ along the negative gradient direction, we have:

$$-\eta\nabla_w \ell(w^*; v^*) = \eta \frac{\|\nabla_w \ell(w^*; v^*)\|_2}{\|w^* - w^0\|_2}(w^* - w^0) = \tilde{\eta}(w^* - w^0), \tag{F.6}$$

where $\tilde{\eta} := \eta \frac{\|\nabla_w \ell(w^*; v^*)\|_2}{\|w^* - w^0\|_2} = \eta \frac{\|\nabla_w \ell(w^*; v^*)\|_2}{\epsilon}$. In this case, the loss function will decrease when we further move $w$ along $-\nabla_w \ell(w^*; v^*)$ with a sufficiently small stepsize $\eta$. In other words, we can apply Descent lemma (details can be seen in Bertsekas et al. [6]) in this local region, i.e.

$$\ell\left(w^* - \eta\nabla_w \ell(w^*; v^*); v^*\right) - \ell(w^*; v^*) \leq -\eta\|\nabla_w \ell(w^*; v^*)\|_2^2. \tag{F.7}$$

As a matter of fact, after further taking a small GD step at $w^*$, $f\left(w^* - \eta\nabla_w \ell(w^*; v^*); v^*\right)$ will be closer to the groundtruth $y$, we will come back to this fact later, it will be used to bound $\|\nabla_w \ell(w^*; v^*)\|_2$.

Now, for any $w \in B_\epsilon(w^0)$, we take the Taylor expansion of $f(w; v^*)$ at $w^0$:

$$f(w; v^*) \quad = \quad f(w^0; v^*) + J(w^0; v^*)(w - w^0) + R(w) \tag{F.8}$$

$$\overset{\text{(a) \& (b)}}{=} \quad J(w^0; v^*)w + R(w), \tag{F.9}$$

$$f_{\text{lin}}(w; v^*) \quad := \quad f(w^0; v^*) + J(w^0; v^*)(w - w^0) \tag{F.10}$$

$$\overset{\text{(a) \& (b)}}{=} \quad J(w^0; v^*)w, \tag{F.11}$$

where $R(w) \in \mathbb{R}^n$ is the residue term of the Taylor expansion:

$$[R(w)]_i = \int_0^1 (w - w^0)^T H_i(w^0 + t(w - w^0))(w - w^0)(1 - t)dt, \quad i = 1, \cdots, n, \tag{F.12}$$

where $H_i(w)$ is the Hessian matrix of $f(w; v^*, x_i)$ at $w$ (for simplicity, we drop the dependence of $v^*$ and $x_i$ in the notation), and $f_{\text{lin}}(w; v^*)$ is a linear approximation of $f(w; v^*)$. In addition, (a) & (b) is due to the fact that $w_j^0$ follows the mirrored LeCun's initialization with $\left(w_{\frac{m}{2}+1}^0, \ldots, w_m^0\right) = \left(w_1^0, \ldots, w_{\frac{m}{2}}^0\right)$, recall the construction of $f(w; v)$ in (6), the hidden output will cancel out with the outer weight, so we have (a):

$$f(w^0; v^*) \stackrel{(6)}{=} \sum_{j=1}^{\frac{m}{2}} v_j^* \left(\sigma(w_j^{0T} x) - \sigma(w_{j+\frac{m}{2}}^{0T} x)\right) = 0. \tag{F.13}$$

Similarly, we have (b):

$$J(w^0; v^*)w^0 = \nabla_w f\left(w^0; x, v^*\right)^T w^0 = \sum_{j=1}^{\frac{m}{2}} v_j^* \left(\sigma'(w_j^{0T} x)w_j^{0T} x - \sigma'(w_{j+\frac{m}{2}}^{0T} x)w_{j+\frac{m}{2}}^{0T} x\right) = 0. \tag{F.14}$$

Based on (F.9), we define the following quantities:

$$\bar{f} := f\left(w^* - \eta\nabla_w\ell(w^*; v^*); v^*\right) \stackrel{(F.9)}{=} J(w^0; v^*)(w^* - \eta\nabla_w\ell(w^*; v^*)) + \bar{R}, \tag{F.15}$$

$$f^* := f\left(w^*; v^*\right) \stackrel{(F.9)}{=} J(w^0; v^*)w^* + R^*, \tag{F.16}$$

where $\bar{R} = R\left(w^* - \eta\nabla_w\ell(w^*; v^*)\right)$, $R^* = R\left(w^*\right)$ as it is introduced in (F.12). Additionally, $\bar{f}$ can be re-written in the form of $f^*$:

$$\bar{f} \stackrel{(F.15)}{=} J(w^0; v^*)(w^* - \eta\nabla_w\ell(w^*; v^*)) + \bar{R} \tag{F.17}$$

$$= J(w^0; v^*)w^* - \eta J(w^0; v^*)\nabla_w\ell(w^*; v^*) + \bar{R} \tag{F.18}$$

$$\stackrel{(F.16)}{=} f^* - R^* - \eta J(w^0; v^*)\nabla_w\ell(w^*; v^*) + \bar{R} \tag{F.19}$$

$$\stackrel{(F.6)}{=} f^* - R^* + \tilde{\eta} J(w^0; v^*)(w^* - w^0) + \bar{R} \tag{F.20}$$

$$\stackrel{(F.14)}{=} f^* - R^* + \tilde{\eta} J(w^0; v^*)(w^*) + \bar{R} \tag{F.21}$$

$$\stackrel{(F.16)}{=} f^* - R^* + \tilde{\eta}(f^* - R^*) + \bar{R} \tag{F.22}$$

$$= (1 + \tilde{\eta})f^* - (1 + \tilde{\eta})R^* + \bar{R}. \tag{F.23}$$

Now, we construct a rough estimator of $\bar{f}$ by merely adding a residue term $(1 + \tilde{\eta})R^*$ on $\bar{f}_{\text{lin}}$, which is $f'$ defined as follows:

$$f' := \bar{f}_{\text{lin}} + (1 + \tilde{\eta})R^* \tag{F.24}$$

$$= f_{\text{lin}}\left(w^* - \eta\nabla_w\ell(w^*; v^*); v^*\right) + (1 + \tilde{\eta})R^* \tag{F.25}$$

$$\stackrel{(F.11)}{=} J(w^0; v^*)(w^* - \eta\nabla_w\ell(w^*; v^*)) + (1 + \tilde{\eta})R^* \tag{F.26}$$

$$= J(w^0; v^*)w^* - \eta J(w^0; v^*)\nabla_w\ell(w^*; v^*) + (1 + \tilde{\eta})R^* \tag{F.27}$$

$$\stackrel{(F.16)}{=} f^* - R^* + \eta J(w^0; v^*)\nabla_w\ell(w^*; v^*) + (1 + \tilde{\eta})R^* \tag{F.28}$$

$$\stackrel{(F.6)}{=} f^* - R^* + \tilde{\eta} J(w^0; v^*)(w^* - w^0) + (1 + \tilde{\eta})R^* \tag{F.29}$$

$$\stackrel{(F.14)}{=} f^* - R^* + \tilde{\eta} J(w^0; v^*)(w^*) + (1 + \tilde{\eta})R^* \tag{F.30}$$

$$\stackrel{(F.16)}{=} f(w^*; v^*) - R^* + \tilde{\eta}(f^* - R^*) + (1 + \tilde{\eta})R^* \tag{F.31}$$

$$= (1 + \tilde{\eta})f^*. \tag{F.32}$$

According to the descent property in (F.7), after taking a very small GD step at $w^*$, the new loss function $\ell \circ \bar{f}$ will be smaller than the old one $\ell \circ f^*$, where $\ell \circ f = \frac{1}{2}\|y - f\|_2^2$. In contrast, we

discuss the change of the loss function from $f^*$ to that of the rough estimator $f'$. The following Lemma F.1 shows that $\ell \circ f' \geq \ell \circ f^*$, different from the fact that $\ell \circ \bar{f} \leq \ell \circ f^*$.

**Lemma F.1.** *[Corresponding to Lemma F.1 in the proof sketch.] Under the background of Theorem 2 and the definition of $f'$ & $f^*$ in (F.32) & (F.16), we have $\ell \circ f' \geq \ell \circ f^*$, i.e., $\|f' - y\|_2^2 \geq \|f^* - y\|_2^2$.*

The proof of Lemma F.1 can be seen in Appendix F.3.

Now, we have

$$\eta \|\nabla_w \ell(w^*; v^*)\|_2^2 \overset{(F.7)}{\leq} \ell(w^*; v^*) - \ell(w^* - \eta \nabla_w \ell(w^*; v^*); v^*) \tag{F.33}$$

$$= \frac{1}{2} \|f^* - y\|_2^2 - \frac{1}{2} \|\bar{f} - y\|_2^2 \tag{F.34}$$

$$\overset{\text{Lemma}F.1}{\leq} \frac{1}{2} \|f' - y\|_2^2 - \frac{1}{2} \|\bar{f} - y\|_2^2 \tag{F.35}$$

$$= \frac{1}{2} \|f' - \bar{f}\|_2 \|f' + \bar{f} - 2y\|_2. \tag{F.36}$$

Next, we bound $\|f' - \bar{f}\|_2 \|f' + \bar{f} - 2y\|_2$ using the following Lemma F.2.

**Lemma F.2.** *[Corresponding to Lemma F.2 in the proof sketch.] Under the background of Theorem 2 and the definition of $f'$ & $\bar{f}$ in (F.32) & (F.15), we have:*

$$\|f' - \bar{f}\|_2 \|f' + \bar{f} - 2y\|_2 \leq \left( \tilde{\eta} \epsilon^2 \lambda_{[0:2]} \sqrt{20n} \right) \left( \sqrt{n} C_y + 3 \left( n \left( \frac{m}{2} L \zeta \epsilon \right)^2 + n C_y^2 \right)^{\frac{1}{2}} \right),$$
$$\tag{F.37}$$

*where $\zeta$ is the constraint for $v$ required in $B(v)$: $v \geq \zeta \mathbf{1}$; $\lambda_{[a:b]} := \max_i \{\lambda_{i[a:b]} | i = 1, \cdots, n\}$, and each $\lambda_{i[a:b]}$ is the maximum eigenvalue of the Hessian matrices $H_i(w^0 + t(w^* - w^0)) := \nabla_w^2 f(w^0 + t(w^* - w^0); v^*, x_i)$ in the interval $t \in [a, b]$.*

The proof of Lemma F.2 can be seen in Appendix F.4. Now, with the help of Lemma F.2, we have:

$$\eta \|\nabla_w \ell(w^*; v^*)\|_2^2 \overset{(F.36)}{\leq} \frac{1}{2} \|f' - \bar{f}\|_2 \|f' + \bar{f} - 2y\|_2 \tag{F.38}$$

$$\overset{(F.37)}{\leq} \frac{1}{2} \left( \tilde{\eta} \epsilon^2 \lambda_{[0:2]} \sqrt{20n} \right) \left( \sqrt{n} C_y + 3 \left( n \left( \frac{m}{2} L \zeta \epsilon \right)^2 + n C_y^2 \right)^{\frac{1}{2}} \right). \tag{F.39}$$

Recall $\eta = \frac{\epsilon}{\|\nabla_w \ell(w^*; v^*)\|_2} \tilde{\eta}$ and $\tilde{\eta}$ is sufficiently small, we have:

$$\|\nabla_w \ell(w^*; v^*)\|_2 \leq \frac{1}{2} \epsilon \left( \lambda_{[0:2]} \sqrt{20n} \right) \left( \sqrt{n} C_y + 3 \left( n \left( \frac{m}{2} L \zeta \epsilon \right)^2 + n C_y^2 \right)^{\frac{1}{2}} \right). \tag{F.40}$$

Since $\zeta$ can be chosen arbitrarily small, we can choose it to be smaller than $\frac{1}{\epsilon}$. That is to say, $\|\nabla_w \ell(w^*; v^*)\|_2 = O(\epsilon)$, so the claim (I) is proved. Note that to be precise, the constant on the right hand side of the above inequality depends on $\lambda_{[0:2]}$, which is a linear function of $v$, and the latter vector may not be upper bounded. Nevertheless, claim (I) is just an intermediate result, and our subsequent derivation will directly use the right hand side of (F.40), which is precise.

Now, we build the relationship between $\|\nabla_w \ell(w^*; v^*)\|_2$ and the loss function. Here, we need to eliminate the dependence of $v$ in the final result: since there is no uniform upper bound for $v$, it can potentially make $\lambda_{[0:2]}$ grow exponentially. To alleviate this issue, we utilize the fact that $f(x; \theta)$ is linear in $v$, so $\lambda_{[0:2]}$ is also linear in $v$, and then we manage to remove the dependence of $v$ in our final result. Specifically, let us define

$$v^*_{\min} := \min_j \{v^*_j \mid j = 1, \cdots, m\}, \quad v^*_{\max} := \max_j \{v^*_j \mid j = 1, \cdots, m\}.$$

So we have

$$\|\nabla_w \ell(w^*; v^*)\|_2^2 = \|J(w^*; v^*)^T (y - f^*)\|_2^2 \overset{(c)}{\geq} v^{*2}_{\min} \cdot \|\tilde{J}(w^*; v^*)^T (y - f^*)\|_2^2 \tag{F.41}$$

$$\geq v^{*2}_{\min} \cdot \tilde{\lambda}^{*2}_{\min} \cdot \|y - f^*\|_2^2, \tag{F.42}$$

where $\tilde{\lambda}^*_{\min}$ is the smallest singular value of $\tilde{J}(w^*; v^*)$, and $\tilde{J}(w^*; v^*)$ is:

$$\tilde{J}(w^*; v^*) := \begin{pmatrix} \sigma'\left(w_1^{*T} x_1\right) x_1^T & \cdots & \sigma'\left(w_m^{*T} x_1\right) x_1^T \\ & \vdots & \\ \sigma'\left(w_1^{*T} x_n\right) x_n^T & \cdots & \sigma'\left(w_m^{*T} x_n\right) x_n^T \end{pmatrix} \in \mathbb{R}^{n \times md}.$$

Here, $(c)$ is straightforward because $\tilde{J}(w^*; v^*)$ is just the simplified version of $J(w^*; v^*)$ by removing all the coefficient $v_j^*$; furthermore, $\tilde{J}(w^*; v^*)$ is full row rank because (i) $J(w^*; v^*)$ is proved to be full rank in the second part of Theorem 1 in Appendix E.2, (ii) $v^*$ is entry-wise non-zero, so $\tilde{\lambda}^*_{\min}$ is strictly positive.

Now, we need to remove the dependence of $v$ in the right hand side of (F.40). Similarly as before, $\lambda_{[a:b]}$ can also be bounded by $v^*_{max}\tilde{\lambda}_{[a:b]}$, where $\tilde{\lambda}_{[a:b]}$ is equal to $\max_i\{\tilde{\lambda}_{i[a:b]} | i = 1, \cdots, n\}$, and each $\tilde{\lambda}_{i[a:b]}$ is the maximum eigenvalue of the Hessian matrices $\tilde{H}_i(w^0 + t(w^* - w^0)) := \nabla_w^2 \tilde{f}(w^0 + t(w^* - w^0)); , x_i)$ in the interval $t \in [a, b]$, and $\tilde{f}(w; , x_i) := \sum_{j=1}^m \sigma(x_i^T w_j)$ is the simplified version of $f(w; v, x_i)$ by removing all the coefficient $v_j$. In conclusion, we have

$$v^*_{\min}\tilde{\lambda}^*_{\min}\|y - f^*\|_2 \leq \|\nabla_w \ell(w^*; v^*)\|_2 \tag{F.43}$$

$$\leq \frac{1}{2}\epsilon\left(\lambda_{[0:2]}\sqrt{20n}\right)\left(\sqrt{n}C_y + 3\left(n\left(\frac{m}{2}L\zeta\epsilon\right)^2 + nC_y^2\right)^{\frac{1}{2}}\right) \tag{F.44}$$

$$\leq \frac{1}{2}\epsilon\left(v^*_{max}\tilde{\lambda}_{[0:2]}\sqrt{20n}\right)\left(\sqrt{n}C_y + 3\left(n\left(\frac{m}{2}L\zeta\epsilon\right)^2 + nC_y^2\right)^{\frac{1}{2}}\right). \tag{F.45}$$

Rearrange and take the square on both sides, we get $\|y - f^*\|_2^2 = O\left(\kappa^2\epsilon^2\right)$, where $\kappa$ is the finite constant in the constraint $B(v)$ of problem (7). So $\ell(\theta^*) = O(\epsilon^2)$, the proof of Theorem 2 is completed.

### F.3 Proof of Lemma F.1

To prove Lemma F.1, we need to discuss the following cases:

(i) When $f^{*T}y \geq 0$, we prove Lemma F.1 by contradiction: when $\tilde{\eta}$ is sufficiently small, suppose $\|(1 + \tilde{\eta})f^* - y\|_2^2 < \|f^* - y\|_2^2$, then $(w^*, v^*)$ is not a KKT point (this case corresponds to the Figure 6 (Middle)). Note that in problem (7), $f^* = \sum_{j=1}^{\frac{m}{2}} v_j^*\left(\sigma(w_j^{*T}x) - \sigma(w_{j+\frac{m}{2}}^{*T}x)\right)$ is linear in $(v_1^*, \cdots, v_{\frac{m}{2}}^*)$, so $(1 + \tilde{\eta})f^* = f(w^*; (1 + \tilde{\eta})v^*)$. That is to say, $\|(1 + \tilde{\eta})f^* - y\|_2^2 < \|f^* - y\|_2^2$ implies $\|f(w^*; (1 + \tilde{\eta})v^*) - y\|_2^2 < \|f^* - y\|_2^2$, which means we can further reduce the loss by changing $v^* \to (1 + \tilde{\eta})v^*$. Since $(1 + \tilde{\eta})v^*$ is still feasible if $v^*$ is feasible, we have a contradiction to the assumption that $(w,^* v^*)$ is a KKT point.

Therefore, in case (i), moving from $f^*$ to $f' = (1+\tilde{\eta})f^*$ will not further reduce the loss (this case corresponds to the Figure 6 (Right)). In other words, we always have $f^{*T}(f^* - y) \geq 0$, this property will also be used in Lemma F.2.

(ii) When $f^{*T}y < 0$, we have

$$\|f^* - y\|_2^2 = \|\Phi(w^*)v^* - y\|_2^2 \tag{F.46}$$

$$= \|\Phi(w^*)v^*\|_2^2 + \|y\|_2^2 - 2(\Phi(w^*)v)^Ty \tag{F.47}$$

$$> \|f^*\|_2^2 + \|y\|_2^2 - 2(-\Phi(w^*)v)^Ty \tag{F.48}$$

$$= \|\Phi(w^*)(-v^*) - y\|_2^2 \tag{F.49}$$

$$= \|-f^* - y\|_2^2, \tag{F.50}$$

where

$$\Phi(w) := \begin{bmatrix} \sigma\left(w_1^T x_1\right), \ldots, \sigma\left(w_m^T x_1\right) \\ \vdots \\ \sigma\left(w_1^T x_n\right), \ldots, \sigma\left(w_m^T x_n\right) \end{bmatrix} \in \mathbb{R}^{n \times m}.$$

Therefore, changing $v^*$ to $-v^*$ will further reduce the loss function (see Figure 6 (Left)). Since $-v^*$ is feasible if $v^*$ is feasible, this is a contradiction to the assumption that $(w^*, v^*)$ is a KKT point. That is to say, we always have case (i): $f^{*T}y \geq 0$.

In conclusion, we always have $\ell \circ f' \geq \ell \circ f^*$, so the proof is completed.

## F.4   Proof of Lemma F.2

Similarly with the residue term (F.12), we define $R \in \mathbb{R}^n$ with each component satisfying $[R]_i := \int_0^{1+\tilde{\eta}}(w-w^0)^T H_i(w^0 + t(w-w^0))(w-w^0)(1-t)dt$, we have:

$$\|f' - \bar{f}\|_2^2 \overset{(F.32)\&(F.15)}{=} \left\|-(1+\tilde{\eta})R^* + \bar{R}\right\|_2^2 \tag{F.51}$$

$$= \left\|(1+\tilde{\eta})R^* - \bar{R}\right\|_2^2 \tag{F.52}$$

$$\overset{(*)}{=} \left\|(1+\tilde{\eta})R^* - (1+\tilde{\eta})^2 R\right\|_2^2 \tag{F.53}$$

$$= \left\|(1+\tilde{\eta})(R^* - R) - (1+\tilde{\eta})\tilde{\eta}R\right\|_2^2 \tag{F.54}$$

$$\leq (1+\tilde{\eta})^2 \sum_{i=1}^n \left(\int_1^{1+\tilde{\eta}}(w-w^0)^T H_i(w^0 + t(w-w^0))(w-w^0)(1-t)dt\right)^2$$

$$+ (1+\tilde{\eta})^2\tilde{\eta}^2 \sum_{i=1}^n \left(\int_0^{1+\tilde{\eta}}(w-w^0)^T H_i(w^0 + t(w-w^0))(w-w^0)(1-t)dt\right)^2$$

$$\leq (1+\tilde{\eta})^2 \sum_{i=1}^n \left(\lambda_{i[1:1+\tilde{\eta}]}\epsilon^2 \int_1^{1+\tilde{\eta}}dt\right)^2 + (1+\tilde{\eta})^2\tilde{\eta}^2 \sum_{i=1}^n \left(\lambda_{i[0:1+\tilde{\eta}]}\epsilon^2 \int_0^{1+\tilde{\eta}}dt\right)^2$$

$$= n(1+\tilde{\eta})^2\tilde{\eta}^2\lambda_{[1:1+\tilde{\eta}]}^2\epsilon^4 + n(1+\tilde{\eta})^4\tilde{\eta}^2\lambda_{[0:1+\tilde{\eta}]}^2\epsilon^4 \tag{F.55}$$

$$= n\tilde{\eta}^2\epsilon^4\left((1+\tilde{\eta})^2\lambda_{[1:1+\tilde{\eta}]}^2 + (1+\tilde{\eta})^4\lambda_{[0:1+\tilde{\eta}]}^2\right) \tag{F.56}$$

$$= n\tilde{\eta}^2\epsilon^4\left(4\lambda_{[1:1+\tilde{\eta}]}^2 + 16\lambda_{[0:1+\tilde{\eta}]}^2\right) \tag{F.57}$$

$$= n\tilde{\eta}^2\epsilon^4\left(20\lambda_{[0:2]}^2\right), \tag{F.58}$$

where the last two inequalities is because of the fact that $\tilde{\eta} \leq 1$ is sufficiently small and $\lambda_{[1:1+\tilde{\eta}]} \leq \lambda_{[0:1+\tilde{\eta}]} \leq \lambda_{[0:2]}$. (*) is due to: for $i = 1, \cdots, n$:

$$[\bar{R}]_i = \int_0^1\left(w-\eta\nabla_w\ell(w^*;v^*)-w^0\right)^T H_i\left(w^0 + t(w-\eta\nabla_w\ell(w^*;v^*)-w^0)\right)\left(w - \eta\nabla_w\ell(w^*;v^*) - w^0\right)(1-t)dt \tag{F.59}$$

$$\overset{(F.6)}{=} (1+\tilde{\eta})^2 \int_0^1\left(w-w^0\right)^T H_i\left(w^0 + (1+\tilde{\eta})t(w-w^0)\right)\left(w-w^0\right)(1-t)dt \tag{F.60}$$

$$= (1+\tilde{\eta})^2 \int_0^{1+\tilde{\eta}}\left(w-w^0\right)^T H_i\left(w^0 + t(w-w^0)\right)\left(w-w^0\right)(1-t)dt \tag{F.61}$$

$$= (1+\tilde{\eta})^2[R]_i. \tag{F.62}$$

Now, we bound $\|f' + \bar{f} - 2y\|_2$, since $(w^*, v^*)$ is a KKT point, it is proved in Lemma F.1 in Appendix F.3 that $f^{*T}y \geq 0$, furthermore, at $w = w^*$, the loss function at $(w^*, v^*)$ should be less or equal to all other feasible points $(w^*, v)$, including $v = \zeta\mathbf{1}$, i.e.,

$$\|f^* - y\|_2^2 = \|f(w^*; v^*) - y\|_2^2 \tag{F.63}$$

$$\leq \|f(w^*; v^*)\|^2 + \|y\|_2^2 \tag{F.64}$$

$$\leq \|f(w^*; \zeta\mathbf{1})\|^2 + \|y\|_2^2 \tag{F.65}$$

$$\overset{(**)}{\leq} n\left(\frac{m}{2}L\zeta\epsilon\right)^2 + \|y\|_2^2 \tag{F.66}$$

$$\leq n\left(\frac{m}{2}L\zeta\epsilon\right)^2 + nC_y^2, \tag{F.67}$$

where the last inequality is because of Assumption 3 (each $y_i \leq C_y$), and (\*\*): is due to

$$\|f(w;\zeta\mathbf{1})\|_2^2 \overset{(6)}{=} \sum_{i=1}^{n} \left( \sum_{j=1}^{\frac{m}{2}} \zeta \left( \sigma(w_j^T x_i) - \sigma(w_{j+\frac{m}{2}}^T x_i) \right) \right)^2 \tag{F.68}$$

$$= \zeta^2 \sum_{i=1}^{n} \left( \sum_{j=1}^{\frac{m}{2}} \left( \sigma(w_j^T x_i) - \sigma(w_{j+\frac{m}{2}}^T x_i) \right) \right)^2 \tag{F.69}$$

$$\overset{\text{Assumption 2}}{\leq} \zeta^2 L^2 \sum_{i=1}^{n} \left( \sum_{j=1}^{\frac{m}{2}} (w_j - w_{j+\frac{m}{2}})^T x_i \right)^2 \tag{F.70}$$

$$\leq \zeta^2 L^2 \sum_{i=1}^{n} \left( \sum_{j=1}^{\frac{m}{2}} \|w_j - w_{j+\frac{m}{2}}\|_2 \|x_i\|_2 \right)^2 \tag{F.71}$$

$$\overset{(***)}{=} \zeta^2 L^2 \sum_{i=1}^{n} \left( \sum_{j=1}^{\frac{m}{2}} \|w_j - w_{j+\frac{m}{2}}\|_2 \right)^2 \tag{F.72}$$

$$\leq n \left( \frac{m}{2} L\zeta\epsilon \right)^2, \tag{F.73}$$

where $(***)$: we assume $\|x_i\|_2 \leq 1$ for $i = 1, \cdots, n$. For general $\|x_i\|_2$, the difference is up to a constant.

Recall $f' = (1 + \tilde{\eta})f^*$, we have

$$\|f' - y\|_2 = \|(1 + \tilde{\eta})f^* - y\|_2 \tag{F.74}$$

$$= \|f^* - y + \tilde{\eta}(f^* - y) + \tilde{\eta}y\|_2 \tag{F.75}$$

$$= \|f^* - y\|_2 + \tilde{\eta}\|f^* - y\|_2 + \tilde{\eta}\|y\|_2 \tag{F.76}$$

$$\overset{(F.67)}{\leq} (1 + \tilde{\eta}) \left( n \left( \frac{m}{2} L\zeta\epsilon \right)^2 + nC_y^2 \right)^{\frac{1}{2}} + \tilde{\eta}\|y\|_2 \tag{F.77}$$

$$\leq (1 + \tilde{\eta}) \left( n \left( \frac{m}{2} L\zeta\epsilon \right)^2 + nC_y^2 \right)^{\frac{1}{2}} + \tilde{\eta}\sqrt{n}C_y \tag{F.78}$$

$$\leq 2 \left( n \left( \frac{m}{2} L\zeta\epsilon \right)^2 + nC_y^2 \right)^{\frac{1}{2}} + \sqrt{n}C_y, \tag{F.79}$$

where the last inequality is because $\tilde{\eta}$ is sufficiently small. Now, combining with the descent property $\|\bar{f} - y\|_2 \leq \|f^* - y\|_2$, we have

$$\|f' + \bar{f} - 2y\|_2 \leq \|f' - y\|_2 + \|\bar{f} - y\|_2 \tag{F.80}$$

$$\leq \|f' - y\|_2 + \|f^* - y\|_2 \tag{F.81}$$

$$\leq 3 \left( n \left( \frac{m}{2} L\zeta\epsilon \right)^2 + nC_y^2 \right)^{\frac{1}{2}} + \sqrt{n}C_y. \tag{F.82}$$

We conclude the proof of Lemma F.2 by combining (F.58) and (F.82).

# G   Extension To Deep Networks

In this section, we discuss how to extend our analysis to deep networks. To do so, we apply the mirrored LeCun's initialization and the constrained formulation (7) to the last two layers and treat the output of the $(L-2)$-th layer as the input features. In the proof of Theorem 1, the expressivity is guaranteed if the inputs $\{x_1, \cdots, x_n\}$ follow a *continuous joint distribution*, which is true under Assumption 3. Fortunately, for deep neural networks with $m_l \geq \frac{2n}{m_{l-1}}$ (where $m_l$ is the width of the $l$-th layer), the outputs of the $(L-2)$-th layer still follow a *continuous joint distribution* under

Assumption 2 and 3, so the expressivity can be shown using the similar technique as Theorem 1. This result is formally stated and proved in the Lemma G.1 below.

**Lemma G.1.** *Given a deep fully-connected neural network with L layers:*

$$f(x;\theta) = w^{(L)}\sigma\left(w^{(L-1)}\ldots\sigma\left(w^{(2)}\sigma\left(w^{(1)}x\right)\right)\right),$$

*where $\sigma(\cdot) : \mathbb{R} \to \mathbb{R}$ is the activation function, $w^{(l)} \in \mathbb{R}^{m_l \times m_{l-1}}$ are the weights, $l = 1,\ldots,L$. Under Assumption 2 and 3, suppose $m_l \geq m_{l+1}$, for $l \leq L - 3$ and $m_{L-1}m_{L-2} \geq 2n$, then at the initialization $\theta^0$ (for $l \leq L - 2$, LeCun's initialization is used; for the last two layers, the mirrored LeCun's initialization is used), for the inputs $\{x_1, \cdots, x_n\}$, the outputs of the $(L-2)$-th layer follow a continuous joint distribution.*

To prove Lemma G.1, we first prove the following lemma:

**Lemma G.2.** *Suppose that $\psi : \mathbb{R}^{k_1} \to \mathbb{R}^{k_2}$ is a analytic mapping and for almost every $u \in \mathbb{R}^{k_1}$, the Jacobian matrix $J(u)$ of $\psi$ w.r.t. $u$ is of full row rank. If $u \in \mathbb{R}^{k_1}$ follows a continuous distribution and $k_1 \geq k_2$, then $\psi(u)$ also follows a continuous distribution.*

*Proof.* Let $Z_0 \subseteq \mathbb{R}^{k_2}$ be a zero measure set in $\mathbb{R}^{k_2}$, We define $S_1(Z_0) = \{u \in \mathbb{R}^{k_1} \mid \psi(u) \in Z_0, J(u)$ is non-singular$\}$. By the definition of $S_1(Z_0)$, any $u \in S_1(Z_0)$ can be written as $u = (u_1^T, u_2^T)^T$, where $u_1 \in \mathbb{R}^{k_2}$ and $J(u_1; u_2)$ is invertible ( $J(u_1; u_2)$ is the $l \times l$ submatrix of $J(u)$, similarly as in Lemma E.1). Then by the Inverse Function Theorem, there exists some ball $\mathcal{B}_{\epsilon(u)}(u) \subseteq \mathbb{R}^{k_1}$ (centered at $u$ with radius $\epsilon(u)$) such that for any $u' = ((u')_1^T, (u')_2^T)^T \in \mathcal{B}_{\epsilon(u)}(u) \cap S_1(Z_0)$, $u'_1 = \tau(u'_2, z')$ , where $z' = \psi(u') \in Z_0$ and $\tau$ is a smooth mapping in a neighborhood $\tilde{Z}_0$ of $(u_2, \psi(u))$. Then for any $u$, there exists a rational point $\bar{u} \in \mathbb{Q}^{k_1}$ and a rational number $\bar{\epsilon}(u) \in \mathbb{Q}$ such that $u \in N(u) := \mathcal{B}_{\bar{\epsilon}(u)}(\bar{u}) \subseteq \mathcal{B}_{\epsilon(u)}(u)$. Since the collection of all open balls with a rational center and a rational radius is a countable set, we let $N_1, N_2, \cdots, N_n, \cdots$ be different $N(u)$ for $u \in S_1(V_0)$. Then $S_1(Z_0) = \cup_{i=1}^{\infty}(N_i \cap S_1(Z_0))$.

We then only need to prove that for any $i$, $N_i \cap S_1(Z_0)$ is of measure zero in $\mathbb{R}^k$. We define the mapping $\tilde{\tau} : \tilde{Z}_0$ as $\tilde{\tau}(u'_2, z') = u'$ if $z' = \psi(u')$. Since $Z_0$ is of measure zero in $\mathbb{R}^{k_2}$, $\tilde{Z}_0$ is measure zero in $\mathbb{R}^{k_1}$. Then because $\tilde{\tau}$ is smooth, the image of $\tilde{\tau}$ of the set $\tilde{V}_0$ is of zero measure in $\mathbb{R}^{k_1}$ (The image of a zero mesure set under a smooth mapping is also measure zero). Notice that $\mathcal{B}_{\bar{\epsilon}(u)}(\bar{u}) \cap S_1(Z_0)$ is contained in the image $\tilde{\tau}(\tilde{Z}_0)$, it is also of zero measure. This finishes the proof. □

Now we prove Lemma G.1. When $l \leq L - 3$, let $u^{(l)} \in \mathbb{R}^{m_l}$ be the output vector of the $l$-th layer. Then $u^{(l+1)} = \psi(w^{(l+1)0}, u^{(l)})$, where $\psi$ is analytic and $w^{(l+1)0}$ is the initial parameter in the $l$-th layer. We now prove that $u^{(l)}$ follows a continuous distribution by induction. When $l = 0$, it is true since $u^{(0)} = (x_1, \cdots, x_n)$ is just the input data, which follows a continuous distribution under Assumption 3. Now suppose $u^{(l)}$ still follows a continuous distribution, we have: (i) since $w^{(l+1)0}$ follows a continuous distribution at the mirrored initialization, $u^{(l)}$ and $w^{(l+1)0}$ follow a continuous joint distribution; (ii) Now, viewing $(w^{(l+1)0}, u^{(l)}) \in \mathbb{R}^{m_l m_{l+1}+m_l}$ as the input of $\psi$, when $m_l \geq m_{l+1}$, $J(w^{(l+1)0}, u^{(l)})$ can be proved to be full row rank w.p.1. using the same technique as Theorem 1. In conclusion, we have $u^{(l+1)}$ follows a continuous distribution by Lemma G.2. Hence, we finish the proof of Lemma G.1.

We further comment a bit on extending the trainability analysis to deep nets. For this part, it requires more detailed analysis because the input feature of the penultimate layer is changing along the training (which is fixed in the shallow case), this topic will be considered as future work. Nevertheless, our idea motivates a better training regime for deep networks, and it is numerically verified in our experiments.

# H Implementation Details & More Experiments

## H.1 Guidance on `PyTorch` Implementation

In this section, we provide sample code to implement the our proposed method for narrow nets training, which can achieve small empirical loss as proved in Theorem 2. We formally state our training regime in Algorithm 2.

---

**Algorithm 2** Our training regime

---

**Set up hyperparameters:**
   Choose a constraint size $\epsilon$, $\zeta$, $\kappa$ and a step size $\eta$.
   Define $B_\epsilon\left(w^0\right) := \left\{w \mid \left\|w - w^0\right\|_F \leq \epsilon\right\}$
   Define $B_{\zeta,\kappa}(v) = \{v | v \geq \zeta\mathbf{1}, \text{and for } \forall v_j, v'_j, v_j/v'_j \leq \kappa\}$.
**Set up the pairwise structure of $v$:**
   Consider $f(x;\theta) = \sum_{i=1}^{\frac{m}{2}} v_j(\sigma(w_j^T x) - \sigma(w_{j+\frac{m}{2}}^T x))$.
**Initialization:**
   Initialize $\theta^0 = (w^0, v^0)$ by the mirrored LeCun's initialization, as shown in Algorithm 1
**Training:**
   Update $v$ via Projected Gradient Descent: $v^{t+1} \leftarrow \mathcal{P}_{B(v)}(v^t - \eta\nabla_v \ell(\theta^t))$.
   Update $w$ via Projected Gradient Descent: $w^{t+1} \leftarrow \mathcal{P}_{B_\epsilon(w^0)}(w^t - \eta\nabla_w \ell(\theta^t))$.
**Until the final epoch $t = T$.**

---

Algorithm 2 can be adopted to deep nets by viewing $w$ as the hidden weights in the penultimate layer (or the final block of ResNet [22] in our computer vision experiments), and view $x$ as the feature outputted by all the previous layers. As shown in Algorithm 2, there are several key ingredients: the pairwise structure of $v$ in (7); the mirrored initialization; and the PGD algorithm. We now demonstrate their implementation in `PyTorch`. Each of them only involves several lines of code changes based on the regular training regime.

**The pairwise structure of $v$ & The Mirrored LeCun's initialization.** .

```python
import torch
import torch.optim as optim
import copy
class ShallowNet(nn.Module):
    def __init__(self, n_input, n_hidden):
        super(ShallowNet, self).__init__()

        self.fc1 = nn.Linear(n_input, n_hidden1,bias=False)
        self.tanh=nn.Tanh()
        self.n_hidden1=n_hidden1
        #Cut down half the width of the output layer
        self.fc2 = nn.Linear(int(n_hidden/2), 1, bias=False)

        #The mirrored initialization
        hidden_half = self.fc1.weight[0:int(n_hidden/2)]
        hidden_layer=torch.cat([hidden_half,hidden_half],dim=0)
        self.fc1.weight = torch.nn.Parameter(hidden_layer)

    def forward(self, x):

        x=self.fc1(x)
        h=self.tanh(x)

        #Keep the pairwise structure of v
        h1=h[:,0:int(self.n_hidden1/2)]
        h2=h[:,int(self.n_hidden1/2):self.n_hidden1]

        x_pred1=self.fc2(h1)
```

```
31         x_pred2=self.fc2(h2)
32         x_pred=x_pred1-x_pred2
33
34         return x_pred
```

To extend the mirrored initialization to deeper nets such as ResNet, we just need to repeat the code line [12 - 14] for every hidden layer, including the BatchNorm layer and the CNNs in the shortcut layers in the Residue block.

**Projected Gradient Descent.** We now demonstrate how to implement PGD. First, we need to copy the parameters at the initialization, will be used for projection.

```
1     # Copy the parameters at the initialization, will be used for
      projection
2     model_initial = copy.deepcopy(model)
```

Then we do the projection after each gradient update.

```
1  def train(model, model_initial, epoch, x,y, optimizer):
2
3      #standard code in regular training
4      clf_criterion=nn.MSELoss()
5      model.train()
6      for i in range(epoch):
7          optimizer.zero_grad()
8          pred=model(x=x)
9          loss = clf_criterion(pred,y)  # calculate current loss
10         loss.backward() # calculate gradient
11         optimizer.step() # update parameters
12
13         # Projection
14         for para,para0 in zip(model.parameters(), model_initial.
      parameters()):
15             #project the hidden layer
16             if para.data.size()[0]==model.n_hidden1:
17                 if torch.norm(para.data - para0.data) > eps:
18                     para.data = para0.data + eps * (para.data - para0.
      data) / torch.norm(para.data - para0.data)
19
20             #project the output layer
21             if para.data.size()[0]==1:
22                 para.data=projectv(para.data)
23
24  def projectv(v):
25      vmax=torch.max(v)
26      argmax=torch.argmax(v)
27      vmin=torch.min(v)
28      argmin=torch.argmin(v)
29      #print(vmax/vmin)
30      #print('vmax',vmax)
31      #print('vmin',vmin)
32      if vmin <0.001:
33          #print('projectv1')
34          v[argmin]=0.001
35      if vmax/vmin>1:
36          v[argmax]=1*vmin
37          #print('projectv2')
38      return v
```

## H.2   Details on Experimental Setup

Our empirical studies are based on the synthetic dataset, MNIST, CIFAR-10, CIFAR-100 and the R-ImageNet datasets. MNIST, CIFAR-10 and CIFAR-100 are licensed under MIT. Imagenet is

licensed under Custom (non-commercial). All the experiments are run on NVIDIA V100 GPU. Here, we introduce our settings on synthetic dataset and R-ImageNet.

- Synthetic datset: For $i = 1 \ldots, 1000$, we independently generate $x_i \in \mathbb{R}^{200}$ from standard independent Gaussian, and normalize it to $\|x_i\|_2 = 1$, and we set the ground truth as $y_i = (1^T x_i)^2$ for $i = 1, \cdots, 1000$. In short, sample size $n = 1000$, input dimension $d = 200$.

- R-ImageNet: This is a specifically constructed "restricted" version of ImageNet, with resolution $224 \times 224$.

  The vanilla ImageNet dataset spans 1000 object classes and contains 1,281,167 training images, 50,000 validation images and 100,000 test images. In our experiments, we use a subset of ImageNet, namely Restricted-ImageNet (R-ImageNet). Similar with [27], we leverage the WordNet [46] hierarchical structure of the dataset such that each class in the R-ImageNet is a superclass category composed of multiple ImageNet classes, noted in Table 2 as "components". For example, the "bird" class of R-ImageNet (both the train and validation parts) is the aggregation of ImageNet-1k classes: [10: 'brambling', 11: 'goldfinch', 12: 'house finch', 13: 'junco', 14: 'indigo bunting'], more details can be seen in Table 2. As a result, there are 20 super classes which contain a total of 190 vanilla ImageNet classes.

Table 2: Classes used in the R-ImageNet dataset. The class ranges are inclusive.

| Class name | Corresponding ImageNet components |
|---|---|
| bird | $[10, 11, 12, 13, 14]$ |
| turtle | $[33, 34, 35, 36, 37]$ |
| lizard | $[42, 43, 44, 45, 46]$ |
| snake | $[60, 61, 62, 63, 64]$ |
| spider | $[72, 73, 74, 75, 76]$ |
| crab | $[118, 119, 120, 121, 122]$ |
| dog | $[205, 206, 207, 208, 209]$ |
| cat | $[281, 282, 283, 284, 285]$ |
| bigcat | $[289, 290, 291, 292, 293]$ |
| beetle | $[302, 303, 304, 305, 306]$ |
| butterfly | $[322, 323, 324, 325, 326]$ |
| monkey | $[371, 372, 373, 374, 375]$ |
| fish | $[393, 394, 395, 396, 397]$ |
| fungus | $[992, 993, 994, 995, 996]$ |
| musical-instrument | $[402, 420, 486, 546, 594]$ |
| sportsball | $[429, 430, 768, 805, 890]$ |
| car-truck | $[609, 656, 717, 734, 817]$ |
| train | $[466, 547, 565, 820, 829]$ |
| clothing | $[474, 617, 834, 841, 869]$ |
| boat | $[403, 510, 554, 625, 628]$ |

In each dataset, the neural network architectures are chosen as follows, all of the following cases satisfy $m \geq \frac{2n}{d}$ or $m_{L-1} \geq \frac{2n}{m_{L-2}}$, where $m_l$ is the width of the $l$-th layer.

- Synthetic dataset: we use 1-hidden-layer neural networks with Tanh activation (except for the last layer, where the output dimension equals 1 and no Tanh applied). We study different widths of the hidden layer among $m = 20, 40, 80, 100, 200, 400, 800, 100, 1200$. All of these cases satisfy $m \geq \frac{2n}{d}$.

- MNIST: we use 2-hidden-layer neural networks with ReLU activation (except for the last layer, where the output dimension equals the number of classes and no activation applied). The input dimension $d = 784$, the width of the 1st layer is fixed with $m_1 = 784$ and we study different widths of the 2nd hidden layer among $m_2 = 64, 128, 256, 512, 784, 1024$. All of these cases satisfy $m_{L-1} \geq \frac{2n}{m_{L-2}}$, where $m_l$ is the width of the $l$-th layer.

- CIFAR-10, CIFAR-100 and R-ImageNet: we use ResNet-18 and we try different number of channels in the 4th block (the i.e., the final block) among $m = 64, 128, 256, 512$ (for

regular ResNet-18, the default number of channels in the 4st block should be 512). All of these cases satisfy $m_{L-1} \geq \frac{2n}{m_{L-2}}$, where $m_l$ is the width of the $l$-th layer.

In each dataset, the setup for algorithms are as follows: as for training regime, we apply the mirrored LeCun's initialization for all the neural network structures mentioned above, and for regular training, we use the regular LeCun's initialization. We use square loss for the synthetic dataset, and multi-class cross entropy loss is used for the rest of the cases. During training, CIFAR-10, CIFAR-100 images are padded with 4 pixels of zeros on all sides, then randomly flipped (horizontally) and cropped. R-ImageNet images are randomly cropped during training and center-cropped during testing. Global mean and standard deviation are computed on all the training pixels and applied to normalize the inputs on each dataset. As it is required in problem (7), in our training regime, the optimization variable for and the output layer is cut off to half, i.e., $v = (v_1, \cdots, v_{\frac{m}{2}})$, the other half is always $-v$; as for the hyperparameters of $B(v)$, we set $\zeta = 0.001$ and $\kappa = 1$. After each iteration, relevant parameters will projected in to their feasible sets. In addition to the general setup above, more customized hyperparameters are listed as follows:

- Synthetic dataset: For both our training regime and the regular training regime, $B_\epsilon(w)$ constraint is added on the weights in the hidden layer with $\epsilon = 0.1, 0.2, 0.4, 0.8, 1, 2, 4, 8, 10, 1000$ ($\epsilon = 1000$ is equivalent to the unconstrained updates for $w$). Gradient Descent with 0.9 momentum is used, and we use different constant learning rates $lr_1, lr_2$ for hidden weights and outer weights, in all cases with different $m$ and $\epsilon$, we grid search learning rate $lr_1$=[1e-4,1e-3,5e-3,1e-2,5e-2,1e-1,5e-1], $lr_2$=[1e-4,1e-3,5e-3,1e-2,5e-2,1e-1,5e-1] and report the best results. The neural network is trained for 200000 iterations.

- MNIST: For our training regime, $B_\epsilon(w)$ constraint is added on the weights in the 2nd layer with $\epsilon = 0.1, 0.2, 0.4, 0.8, 1, 2, 4, 8, 10, 1000$ ($\epsilon = 1000$ is equivalent to the unconstrained updates for $w$). In each case, we either use Adam with 0.001 initial learning rate and 1e-4 weight decay, or Stochastic Gradient Descent (SGD) with 0.01 initial learning rate, 0.9 momentum and 5e-4 weight decay, and we report the best results. For both training regime and the regular training regime, we use cosine annealing learning rate scheduling [43] with $T_{\max}$=number of epochs , and the neural network is trained for 200 epochs and batch size of 64 is used.

- CIFAR-10, CIFAR-100: For our training regime , $B_\epsilon(w^0)$ constraint is added on the 4-th block with different constraint size among $\epsilon = 0.1, 0.2, 0.4, 0.8, 1, 2, 4, 8, 10, 1000$ ($\epsilon = 1000$ is equivalent to the unconstrained updates for $w$). For both our training regime and the regular training regime, SGD with 0.1 initial learning rate, 0.9 momentum and 5e-4 weight decay is used, and we use cosine annealing learning rate scheduling with $T_{max}$=number of epochs, and the neural network is trained for 600 epochs and batch size of 128 is used.

- R-ImageNet: For our training regime, $B_\epsilon(w^0)$ constraint is added on the 4th block (i.e., the final block) with different constraint size among $\epsilon = 0.01, 0.1, 1, 1000$ ($\epsilon = 1000$ is equivalent to the unconstrained updates for $w$), and the weights in the 4th block are projected into the constraint after each mini-batch iteration. For both our training regime and the regular training regime, SGD with 0.1 initial learning rate, 0.9 momentum and 5e-4 weight decay is used, we use a stage-wise constant learning rate scheduling with a multiplicative factor of 0.1 on epoch 30, 60 and 90. The neural network is trained for 90 epochs and batch size of 256 is used.

## H.3 Test Accuracy on MNIST

Figure 7 shows the test accuracy in our training regime vs regular training regime in MNIST. With proper choice of $\epsilon$, our training regime leads to higher test accuracy.

## H.4 Test Accuracy on CIFAR-10 & CIFAR-100

In CIFAR-10 and CIFAR-100 dataset, our training regime and regular training regime have similar performance (see Figure 8). In several cases, our training regime leads to higher test accuracy. Here, regular training will not fail when we reduce the width of 4-th block, perhaps this is due to the

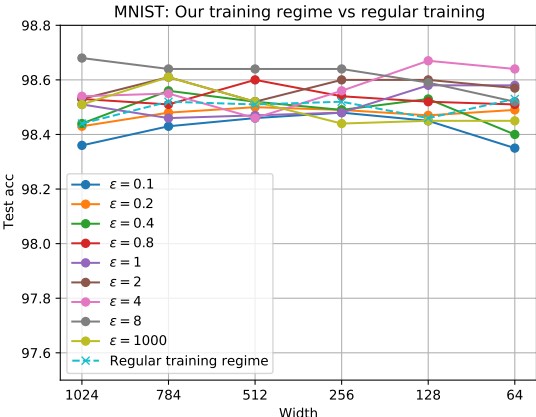

Figure 7: MNIST: test accuracy in our training regime with different $\epsilon$ vs regular training regime. In x-axis, width stands for width of 2nd layer (we use 2-hidden-layer neural nets here).

strong expressivity of ResNet-18. In comparison, on a more complicated dataset such as R-ImageNet, narrowing ResNet-18 will jeopardize the regular training (as illustrated in Section 5 and the following subsection).

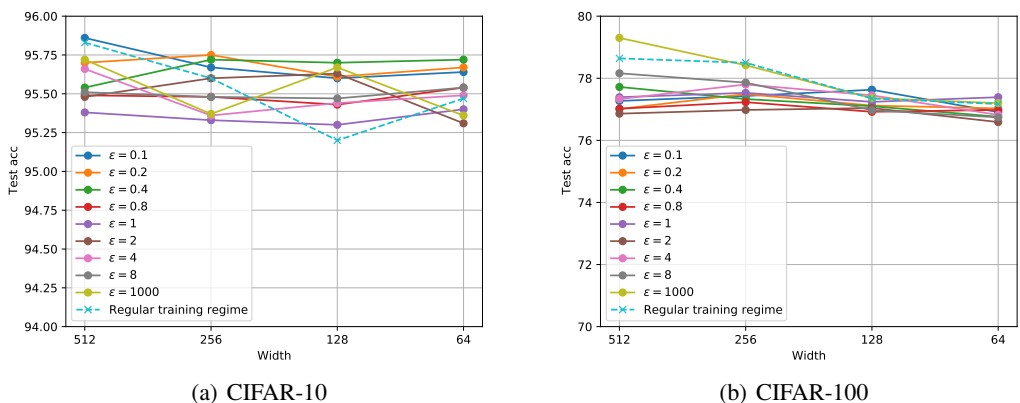

(a) CIFAR-10                                                 (b) CIFAR-100

Figure 8: CIFAR-10 & CIFAR-100: test accuracy in our training regime with different $\epsilon$ vs regular training regime. In x-axis, width stands for the number of channels in the final CNN block of ResNet-18.

### H.5 Test Accuracy on R-ImageNet

In this subsection, Figure 9 is the same figure as Figure 5 in the full paper, but with 90% confidence error bars (based on 5 seeds). Besides, Figure 10 shows a selected result of Loss & Accuracy per epoch in our training regime with $\epsilon = 0.04$ & regular training regime (here, we present the early-stopped results). As a result, our training regime can reduce the number of parameters in the 4-th block of ResNet-18 by up to 94% while maintaining competitive test accuracy, especially when $\epsilon$ is small. In comparison, regular SGD does not perform well in narrow cases.

### H.6 Results on Random-Labeled CIFAR-10

Since the main claim of our work is about memorization of *any* labels, we further explore the performance of our training regime even when the labels are not the correct ones. To do so, we further carry out experiments on the random-labeled CIFAR-10, where all the labels are randomly

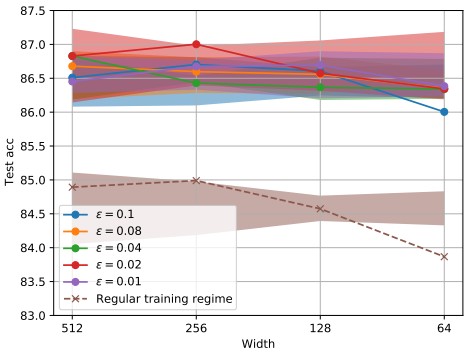

Figure 9: R-ImageNet: test accuracy under our training regime with different $\epsilon$ vs regular training regime. In x-axis, width stands for the number of channels in the final CNN block of ResNet-18. The solid & dotted lines are averaged results over 5 seeds, the shaded areas indicates the 90% confidence intervals.

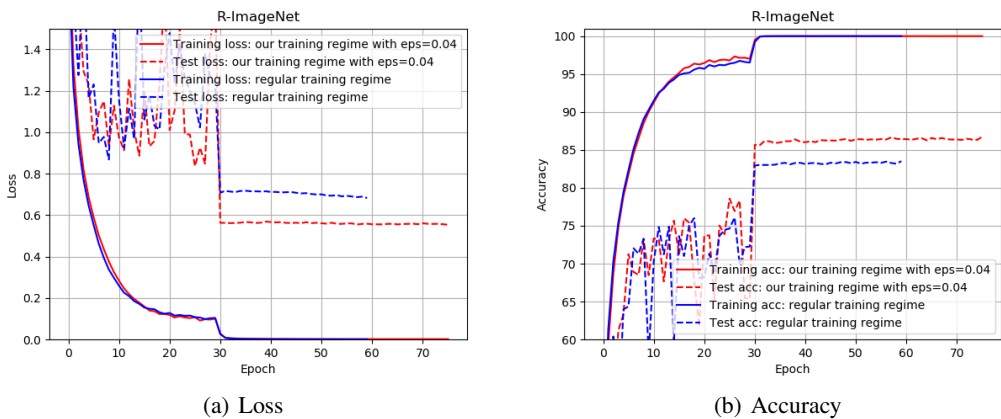

(a) Loss            (b) Accuracy

Figure 10: R-ImageNet (selected): loss & Accuracy per epoch in our training regime with $\epsilon = 0.04$ & regular training regime.

shuffled (same as in Zhang et al. [75]). We train a 1-hidden-layer network with width =1024, 2048, 4096 (smaller than $n$=50000) on the random-labeled CIFAR10 dataset. The hyperparameters in our constrained training regime (7) are $\epsilon = 10$, $\kappa = 1$ and initial learning rate =0.1. We use a stage-wise constant learning rate scheduling with a multiplicative factor of 0.1 on epoch 150, 225, 450. The result is shown in Table 3: after 1000 epochs, we can achieve more than 99% train accuracy, almost perfectly fit the random labels. Note that even though ReLU does not fall into our analysis framework, it works a bit better than Tanh. Extending our results to ReLU activation would be our intriguing future work.

Table 3: Results on the random-labeled CIFAR-10

| Width | Epoch | Activation | Train acc | Test acc |
|-------|-------|------------|-----------|----------|
| 1024 | 1000 | ReLU | 0.9931 | 0.1011 |
| 2048 | 1000 | ReLU | 0.9984 | 0.1022 |
| 4096 | 1000 | ReLU | 0.9998 | 0.0962 |
| 1024 | 1000 | Tanh | 0.9872 | 0.0991 |
| 2048 | 1000 | Tanh | 0.9927 | 0.1024 |
| 4096 | 1000 | Tanh | 0.9938 | 0.0962 |