# OpenReview forum: "When Expressivity Meets Trainability: Fewer than $n$ Neurons Can Work"
_NeurIPS.cc/2021/Conference — NeurIPS 2021 Poster_

### Official Review · Reviewer_wktj · 2021-06-28

**Rating:** 6
**Confidence:** 4

**Summary:**

The paper studies the expressivity and trainability of shallow 2-layer neural networks. First, in terms of expressivity it is shown, under certain assumptions, that for a 2-layer network, it is enough to have 2n/d neurons to memorize n data points in d dimensions. In addition, it is shown that in the close neighborhood of the initialization, every stationary point w.p 1 is a global minimum. Next, it is shown that for a certain constraint setting, every KKT point is an approximated global minimum. Finally, several experiments are shown to validate the theoretical results.

**Limitations And Societal Impact:**

The authors adequately addressed the limitations and potential negative societal impact of their work

**Main Review:**

I think the questions of how many neurons are required for memorization, and whether neural networks actually converge to a global minimum are very interesting. On the other hand, there are two major issues that significantly impair the contribution and novelty of this paper:

1) There are already several papers showing that only O(n/d) neurons are required for memorizing n data points in d dimensions, which are not mentioned in this paper (e.g. see Daniely 2019 https://arxiv.org/abs/1911.09873, Bubeck et al. 2020 https://arxiv.org/abs/2006.02855, and the references therein). In Bubeck et al. the authors also show a convergence result to the global minimum. In light of this, it is very hard to evaluate the novelty of this paper. I suggest the author to cite these papers and highlight the novel part of their work. Specifically, why the results here are different from previous results which also show similar memorization and convergence guarantees.

2) There is a major issue with the proofs. In Theorem 1 the authors claim to show that “every stationary point \theta^* is a global min…”. On the other hand, in the proofs, the authors show something completely different. Namely, that w.p 1 over the initialization, the Jacobian of the loss is invertible. Next, it is shown that outside of a manifold of zero measure (where the Jacobian is not invertible) there are no bad global minima or saddle points. I don’t see how this theorem is helpful since there still might be many bad local minima close to the initialization. Even a single bad local minimum (which as a single point has measure zero) can be disastrous for the optimization process. From my understanding, what the authors actually show is that all the bad local minima and saddle points are located in a zero-measure manifold where the Jacobian is not invertible. This in itself is not surprising, and I don’t currently see how this observation can help with analyzing the optimization process.

This issue persists to Theorem 2, where the authors show that under certain constraints every KKT point w.p 1 is an approximated global minimum. Again, in the proof, the authors seem to assume that the only bad local minima are in the boundary. On the other hand, for any \epsilon >0 there can be bad global minima in the interior of the considered set.

In order to claim that the only bad local minima are in the boundary of the constraint set, the authors should prove that either (1) The Jacobian is invertible at *all* the points of this set. Not w.p 1, or outside of a zero measure set as even a single bad local minima ruin this argument; or (2) That if the Jacobian is not invertible at some point inside this set, then this point is not a bad local minimum (or saddle point). The fact that the set of bad local minima is of measure zero does not say anything about whether GD will converge to these points.


The experimental part is nice, and many datasets are considered. To further improve it, I suggest also conducting experiments on these datasets but with randomized labels. Since the claim of the paper is about memorization of *any* labels, a more convincing experimental argument is that even when the labels are not the correct ones, neural networks with sufficient width can memorize them (this flavor of experiments also appears in Zhang et al. 2017). Also, the text in figure 5 (and also other figures) is very small and makes it difficult to read.

To conclude, the paper studies an interesting question, but there are major issues that make it difficult to assess the contribution of the paper. I will be happy to see the author’s response and will consider changing my score accordingly.


----------------------------Post rebuttal comment----------------------------

I thank the authors for their feedback. After reading the author's response and rereading the proof in their paper I decided to raise my score to 6. Here are my explanations:

Regarding my second remark about the correctness of their proof. I was admittingly mistaken and I am now convinced that what I thought was wrong is actually true. So, I drop my second claim. The proof seems sound.

Regarding the similarity to Daniely's and Bubeck et al. papers. I agree that there are differences between the current paper and these two papers, both in the results and techniques. On the other hand, all three papers essentially claim the same thing - memorization using O(n/d) parameters and *some* optimization guarantees of convergence to a global minimum. I think the authors should have done a better job comparing their work to previous works, probably dedicating a paragraph/subsection to the comparison to these two papers.

For this reason, I think this paper is only slightly above the threshold.



**Time Spent Reviewing:**

5

---

> ### Author Response · Authors · 2021-08-09
> **To Reviewer wktj: Part I: Related Works**
>
> Thanks for your insightful comments! Below, we provide detailed responses to your concerns.
>
> ---
> ---
>
> > *Q1: Related work [R1]: Bubeck, Sébastien, Ronen Eldan, Yin Tat Lee, and Dan Mikulincer. "Network size and weights size for memorization with two-layers neural networks." arXiv preprint arXiv:2006.02855 (2020).*
>
> Thanks for pointing out this reference. We will add the reference and discussions in the revised version.
>
> [R1] studies how many neurons are required for memorizing a finite dataset by 1-hidden-layer networks. They provide two results:  (note that their width $k$ is our $m$, we will use our notation below)
>
> i) There exists a network with width $m \geq \frac{4n}{d}$ that can memorize $n$ input-label pairs (their Proposition 4).
>
> ii)  To achieve memorization up to error $\epsilon$, they propose a new training procedure with width $m=O(\left(\frac{n}{d} \frac{\log (1 / \epsilon)}{\epsilon}\right))$ (Their thm 5).
>
> The major differences between their work and ours are as follows:
>
> 1. Their Proposition 4 is about the **existence** of a network that memorized n input-label pairs using width m=4n/d. However, they didn’t provide an algorithm to find the 0-loss solution. This is a pure ‘existence’ result.
>     In contrast, we provided an algorithm and show experimental results.
>
> 2. In their Thm 5, **the required width depends on the precision $\epsilon$**. More specifically, they need $O\left(\frac{n}{d} \frac{\log (1 / \epsilon)}{\epsilon}\right)$ neurons. When $\epsilon =1/n$, they need $O(n^2/d)$ neurons, which is much larger than the number of samples if $n >> d$.
>   In contrast, in our work, the required number of neurons is just m $\geq$ 2n/d, which is independent of $\epsilon$.
>
> 3.  They don't have experiments for their training procedure. We empirically show that our algorithm can improve the performance of narrow networks.
>
> Remark: We explain how to compute the width requirement of their Thm 5. In their Thm 5, the width bound is $m= \tilde{O} \left(\gamma^{2} \frac{\log (1 / \varepsilon)}{\varepsilon} n \right)$, where  $\gamma$ describes the discrepancy of the input data. As they discussed in the paragraph after their Theorem 5,  $\gamma \approx \widetilde{O}\left(\frac{1}{\sqrt{d}}\right)$ under their setting. Plugging in $\gamma$, we get the desired bound.
>
>
> ---
> ---
>
>
> > *Q2: Related work [R2]: Daniely, Amit. "Neural Networks Learning and Memorization with (almost) no Over-Parameterization." NeurIPS 2020. https://proceedings.neurips.cc/paper/2020/file/662a2e96162905620397b19c9d249781-Supplemental.pdf *
>
> Thanks for pointing out this reference. We will add the reference and discussions in the revised version.
>
> [R2] proved two results on memorizing $n(1-\epsilon)$ random input-binary-label pairs by 1-hidden-layer networks, with different width requirements:
>
>  i)Thm 5 requires width $\tilde{O}\left(\frac{n}{ \epsilon^{2}}\right)$.
>
>  ii)Thm 7 claimed to require width $\tilde{O}(n/d)$.
>
> The differences between their results and ours are as follows.
>
> **Summary of major differences**
>
> **Difference 1:** our required width is smaller than $n$ (when $d >2$), and their required width is often much larger than n. In fact, their width is actually at least $O(n^2)$ or larger for fixed $d$, though they claimed an order of $\tilde{O}(n/d)$. We will elaborate below.
>
> **Difference 2:** They analyze SGD, and we analyze a constrained optimization problem and projected SGD. This may be the reason why we can get a stronger bound on width.
>
> **Difference 3:** They don't have experiments. We empirically show that our algorithm can improve the performance of narrow networks.
>
> **Elaborating Difference 1 with [R2]**
>
> i) Their Thm 5 provides a width bound dependent on precision $\epsilon$. Similar to [R1], the required width grows as $\epsilon$ grows. The authors wrote before Thm 7 on page 8 that if $\epsilon < 1/n$ then the required parameter number is $O(n^3)$; see [R2, Line 282]. This is why they provided another result Thm 7.
>
> ii) For Thm 7, the **required width contains an additional '$(\log(d \log n)^{\log n}$' term in $\tilde{O}$**; see detailed computation in Remark 1 and more discussions in Remark 2 below. This extra factor is larger than
>    $$  \max \\{   \log(d)^{\log n} ,  [\log(\log n)]^{\log n}     \\}  \triangleq \max \\{ B_1, B_2 \\}.  $$
>
>  This is NOT a constant factor or a log-factor, but more like a polynomial factor on $n$, as explained below.
>
>  (ii.a) We consider the first bound $B_1$ for fixed input dimension $d$ for two cases.
>
> --For $d$ satisfying $\log d > 2.7 $ (i.e. $d>15$), their required width is actually $ O( \frac{n^2}{d} ) $. This is an order of magnitude larger than our bound $O(n/d)$.
>
> --For $d$ satisfying $\log d > 2.7^2 $ (i.e. $d>1395$), the required width is actually $O( \frac{ n^3}{ d}  )$, two orders of magnitude larger than $O(n/d)$.
>
>    (ii.b) Now we briefly discuss $B_2$ in Thm 7. Similar to the above discussion, when $n > 2.7^{{2.7}^{2.7}} \approx 2.2 \times 10^6$, we have $(\log(\log ( n ))^{\log n} > n$. As a result, the required width is at least $ O( \frac{n^2}{d} ) $, no matter how large $d$ is.
>
>    (ii.c) It is not hard to argue that their required width can be larger than $n^{k}$ for any fixed integer $k$. In other words, their network can be significantly overparameterized in some cases.
>
> Therefore, their **required width $\tilde{O}(n/d)$ is not $O(n/d)$, but can be larger than $ O( \frac{n^2}{d} ) $ or even $O( \frac{ n^3}{ d}  )$**.
>
> **Remark 1**: We explain how to compute the width requirement of their Thm 7. Their statement only mentioned $\tilde{O}(n/d)$ but not the exact expression. To find the precise width bound, we tracked the proofs as follows (note that their width $q$ is our $m$ and their sample size $m$ is our $n$, and we will use our notation below):
>
> Step 1: Check Theorem 16 in Section 4.3 'proof of theorem 6' (Page 16, Line 477). To get the desired bound, they need to make sure the 3rd term of the 1st equation in their Thereom 16 is no more than 1, i.e., the denominator is larger than the nominator. As a result, their required width is $ m >= \frac{n}{d} (\log(  d / \delta ))^{(c^\prime+2)}$. Here $\delta = 1/\log n$ and $c^\prime \ge 4c+2$, where $c = \log n/ \log d$, as specified in the next two steps.
>
> Step 2: Identify $c= \log n/\log d$. This can be derived from Line 459: since n/d = d^{c-1}, we have $c = \log n/\log d$. This definition of $c$ first appeared in Line 270 of Sec. 3.2: they assume the number of samples is $n = d^c$. Realizing that $c$ is not a constant, we obtain the extra factor $B_1$ discussed above (we simplify $c$ to $\log n$ when we discuss $B_1$ since for fixed $d$, $c $ grows as $\log n$).
>
> Step 3: Identify $\delta = 1/\log n$. This is specified in their Line 479. Realizing $\delta$ is not a constant, we obtain the extra factor $B_2$ discussed above.
>
> **Remark 2**: The main reason why there is a gap between the claimed $\tilde{O}(n/d)$ width and the actual width is the following: [R2] treated $c$ and $\delta$ as constants, but they actually depend on $d$ and $ n $. It is NOT rigorous to hide factors that are dependent on $n , d$ in the expression $\tilde{O}(n/d)$.
>
> **Remark 3**: On CIFAR10 dataset, $n=50000, d=3072$, thus their required width $m\geq  58290499136$ >> $n$; the calculation is shown below (by numpy code). In comparison, our required width is only $m\geq 2n/d$, which is smaller than $n=50000$.
>
> import numpy as np
>
> d=3072
>
> n=50000
>
> c=np.log(n)/np.log(d)
>
> cprime= 4*c+2
>
> width=int((n/d)*(np.log(d*np.log(n))**(cprime+2)))
>
> print(width) # width 58290499136
>
> **Other differences with [R2]**:
> 1. They focus on memorizing the $1-\epsilon$ fraction of the dataset,
> instead of memorizing the whole dataset as we do.
>
> 2. They consider binary $\\{+1,-1\\}$ dataset, while our results apply to arbitrary labels.

---

> ### Author Response · Authors · 2021-08-10
> **To Reviewer wktj: Part II**
>
> > *Q3: Major issues for the proof of both thm1 and thm2.*
>
>  Thanks a lot for sharing your understanding! However, what you stated are NOT what we did. We would like to clarify as follows.
>
>  Your summary of our result is "every stationary point $\theta^*$ is a global min…". Your summary of our proof's essence is "all the bad local minima and saddle points are located in a zero-measure manifold where the Jacobian is not invertible" and then claim "this does not help optimization" and "this is not surprising". Let us clarify one by one.
>
>  *First, the main result*. There are three possible claims one could make, in generic optimization language.
>
>  Claim 1: Sub-optimal stationary points lie in a zero-measure set.
>
>  Claim 2: In a ball, sub-optimal stationary points do not exist.
>
>  Claim 3: For a constrained optimization problem, sub-optimal KKT points do not exist.
>
>  The reviewer thought we proved Claim 1, **but we actually proved Claim 2 (in Thm 1)**. We did notice that Claim 1 is not helpful for optimization because, as the reviewer pointed out, "the fact that the set of bad local minima is of measure zero does not say anything about whether GD will converge to these points." and "one stationary point can ruin the optimization". This is why we proved Claim 2, which restricts the attention to local landscape so that a stronger conclusion "every stationary point in a ball is global-min" can be proved.
>
>  We further notice that Claim 2 is not enough for the trainability guarantee since it is not easy to ensure GD stays in the ball. Many NTK papers ensure the small movement of weights by using a large width. We do not want to resort to large width, so we directly add a ball constraint to avoid the iterates from moving far away. Now the challenge is to show the KKT points are good, which is what we show in Theorem 2. **Thus the major result we prove is Claim 3 (in Thm 2)**.
>
> *Second, the proof.* The essence of our proof is not just "all the bad stationary points are located in a zero-measure manifold where the Jacobian is not invertible", but **two additional things**:
>    i) stationary point with invertible Jacobian exists (expressiveness);
>    ii) KKT points of a modified problem are good.
> These proof techniques together lead to a result that is useful for optimization.
>
> > *Q4: experiments on the random-labeled dataset.*
>
> Thanks for your great suggestion! The experiment on the random-labeled dataset matches the theme of our paper. Similar to Zhang et al. 2017, we train a 1-hidden-layer network with width $m$=1024, 2048, 4096 ($m<n$=50000) on the fully-random-labeled CIFAR10 dataset. The hyperparameters in our constrained training regime (cf. eq (5)) are eps=10, kappa=1, learning rate=0.1. The result is as follows: we can achieve more than 0.99 train acc, almost perfectly fit the random label in 1000 epochs.
>
> | Width | Epoch | Activation | Train acc | Test acc |
> |-------|-------|------------|-----------|----------|
> | 1024  | 1000  | relu       | 0.9931    | 0.1011   |
> | 2048  | 1000  | relu       | 0.9984    | 0.1022   |
> | 4096  | 1000  | relu       | 0.9998    | 0.0962   |
> | 1024  | 1000  | tanh       | 0.9872    | 0.0991   |
> | 2048  | 1000  | tanh       | 0.9927    | 0.1024   |
> | 4096  | 1000  | tanh       | 0.9938    | 0.0962   |
>
>  Note that even though relu does not fall into our analysis framework, it works a bit better than tanh. Extending our results to relu activation would be our intriguing future work.
>
> > *Q5: The text in the figures is too small.*
>
> Thanks a lot for your suggestion! We have replotted the figures according to your suggestion. Since no revision of the submission script is allowed during the review session, it will appear in the future version.

---

> ### Author Response · Authors · 2021-08-20
> **To Reviewer wktj: Post rebuttal comment**
>
> We’d like to thank the reviewer for careful reading of our response and re-checking our proof. We are grateful that the reviewer spends much time reading our response & proof.  Following your advice, we will add an additional paragraph to compare our work to Daniely 2019 & Bubeck et al. 2020. We will also re-polish the theorem statement for more clear and better presentation.
>
> Thanks again for your follow-up feedback. We really appreciate such high-quality review experience.

---

### Official Review · Reviewer_xfuA · 2021-07-07

**Rating:** 6
**Confidence:** 4

**Summary:**

This paper studies the expressivity and trainability of two-layer (one-hidden-layer) neural networks with the number $m$ of neurons fewer than the number $n$ of samples. Specifically, it is shown that if $m\geq 2n/d$, then with probability one, there exists at least one global minimum with zero training loss. Moreover, there is no local minimum or saddle around such global minimum. The authors also proposed a constrained minimization problem whose KKT condition implies the near global optimality for the unconstrained problem. Numerical tests show that the proposed training regime performs better than the regular training regime.

**Limitations And Societal Impact:**

yes

**Main Review:**

The paper does an excellent job in analyzing the expressivity and trainability of shallow neural networks in the regime that the network width is fewer than the sample size. The results presented in the paper complements the recent theoretical studies on the convergence of neural network training in the overparameterized regime (e.g. Neural Tangent Kernel regime).  From my perspective, the paper is among the top 30% of all submissions. I only have the following minor comments.

*My major comment is about the generalization. The main results claim that there exists a shall neural network that attains zero empirical loss, but the generalization performance of such network was not discussed. To guarantee a zero loss, the parameters $v$ are used to scale the initial value $v_0$ to match the output $y$. It is possible that $v$ can be extremely large depending on the values of $v_0$ and $y$. Large values of $v$ may induce large generalization error. In Section 3.3, the author rules out the large values of $v$ by requiring that the ratio of $v_i$s are bounded by $\kappa$. However, I do not see how large $\kappa$ is and how the final convergence guarantee (equation 6) depends on $\kappa$. It would be nice if the authors could make this more precise.
*The main results were proved for analytical activation functions. Can the authors comment on to which extent the result can be extended to non-smooth activations, such as ReLU or powers of ReLU?
*Some important relevant references on convergence of GD for shall networks are missing:

[1] S. Oymak and M. Soltanolkotabi, Toward moderate overparameterization: Global convergence guarantees for training shallow neural networks, IEEE Journal on Selected Areas in Information Theory 1 (1), 84-105

[2] S. Oymak and M. Soltanolkotabi, Overparameterized nonlinear learning: Gradient descent takes the shortest path?, ICML 2019.

After receiving the feedbacks of the other reviewers, I find that the paper share substantial similarities to some earlier works. I have now changed score.


**Time Spent Reviewing:**

48 hours

---

> ### Author Response · Authors · 2021-08-10
> **To Reviewer xfuA**
>
> Thanks for your encouraging and supportive feedback!  We provide our responses to your comments below.
>
> > *Q1: Large values of v may induce large generalization error*
>
> Thanks for pointing it out.  Indeed, to reach 0 training loss, the optimal $v$ needs to be rather large, especially when $\epsilon$ is small. However, it is not a big issue for generalization. This is because the generalization error is related to the Lipschitz continuity of the neural network, as we briefly argue below. In our eq(4), the neural network can be written as:
>
> $$f(x ; w, v)=\sum_{j=1}^{\frac{m}{2}} v_{j}(\sigma(w_{j}^{T} x)-\sigma(w_{j+\frac{m}{2}}^{T} x))$$
>
> Its Lipschitz constant is bounded by
> $\mathcal{O}(\\|v\\| \max_i\\|w_{2i-1}-w_{2i}\\|)=\mathcal{O}(\\|v\\| \epsilon)$.
>
>  As shown in Appendix A.1,  $\\|v\\|$ needs to be scaled up to fit the labels. Roughly speaking, $\\|v\\|$ grows in the order $1/ \epsilon$. In addition, $\\|w_{2i-1}-w_{2i}\\|$ will never exceed $\epsilon$ due to the constraint. Therefore, the Lipschitz constant of $f(x;w,v)$ is roughly $\mathcal{O}(1)$ and remains bounded. Nevertheless, it is an informal analysis and the more rigorous analysis of the generalization ability will be considered as future work.
>
> Despite the lack of theory, we can always evaluate the generalization performance via numerical experiments. According to our experiments on various vision datasets (in Section 4.2 & Appendix D), our training regime brings out competitive or even better generalization performance than the regular training.
>
> > *Q2: how large kappa is and how the final convergence guarantee (equation 6) depends on kappa. It would be nice if the authors could make this more precise. *
>
> Thanks for your comments. In Appendix B, page 21, we show that equation 6 is in the order of O(kappa^2 epsilon^2).
>
> As for the value of kappa, we chose it to be very large (like kappa=10 or 100) in some of our early-designed experiments.  But later on, we find out that vmax/vmin never exceed 1 during the training (within 0.5 and 0.9, it will be reported in the future version). So later we re-set the kappa=1 and it reaches the same performance. In conclusion, kappa=1 is enough to re-implement all the experiments we reported. In this sense, the kappa term will not jeopardize the quality of our results in thm2.
>
> > *Q3:  To which extent the result can be extended to non-smooth activations, such as ReLU or powers of ReLU?*
>
> We will comment on the possibility. A main reason that we analyze smooth activation is that we follow the approach of Li et al.'2018 [R1] to show the full-rankness of Jacobian. There exist papers that show the full-rankness of Jacobian for ReLU activation, e.g., Du et al.'18 [R2].
>
> Nevertheless, we choose the analysis of Li et al.'2018 since we find a simple way to extend their proof (they prove full-rankness of Jacobian of wide nets, and we extend the proof to prove the full-rankness of Jacobian of narrow nets). We suspect that combining some techniques of this paper and the proof of, say, Du et al.'18, can lead to results for narrow-nets that cover ReLU.
>
> [R1] Li, Dawei, Tian Ding, and Ruoyu Sun. "On the benefit of width for neural networks: Disappearance of bad basins." arXiv preprint arXiv:1812.11039 (2018).
>
> [R2] Du, Simon S., et al. "Gradient descent provably optimizes over-parameterized neural networks." arXiv preprint arXiv:1810.02054 (2018).
>
>
> > *Q4: Some important relevant references on the convergence of GD for shall networks are missing*
>
> Thanks for pointing these related works. We will include them in Section 1. In particular, we'd like to thank the reviewer for pointing out the very interesting work 'Overparameterized nonlinear learning: Gradient descent takes the shortest path?', which seems related to our topic.

---

> > ### Comment · Reviewer_xfuA · 2021-08-18
> > **Decrease my score**
> >
> > I want to thank the authors for the feedbacks. It seems that the paper share some similarity to the paper  "Neural networks learning and memorization with (almost) no over-parameterization" and "Network size and weights size for memorization with two-layers neural networks".   For this reason, I decrease my score from 8 to 6.

---

> > > ### Author Response · Authors · 2021-08-19
> > > **Response to the comment 'Decrease my score'**
> > >
> > > Thank you for your feedback. We'd like to briefly explain the **significant differences** between our work and theirs, in case you did not get a chance to read our response 'To reviewer wktj: Part I: Related Works' (which contains a detailed comparison; you may find it by searching 'Part I: Related Works' on this webpage).
> > >
> > >    **Main difference 1**: For both works, the required width for successful training can be $O(n^2/d)$ or larger.
> > >
> > > More specifically, the width in [R1] is about $\Omega(n/d \frac{1}{\epsilon})$, and the width in [R2] is at least $ \Omega(n/d (\log d)^{\log n})$. For certain parameters (of $\epsilon$ annd $d$), these bounds can be $\Omega(n^2/d)$ or even $\Omega(n^3/d)$.
> > >
> > > In contrast, we only require $2n/d$ neurons.
> > >
> > >   **Main difference 2**: Both works only analyze SGD, but it is known that regular SGD cannot train narrow networks well. We propose a new method to train narrow nets. Experiments show that our method significantly outperforms SGD for narrow nets.
> > >
> > > Finally, we believe these two differences are related. We conjecture **it is impossible to prove SGD converges to global-min for narrow nets** (if without strong assumptions). As a result, they cannot analyze the SGD behavior with width exactly $\Omega(2n/d)$. We suspect an algorithmic change is needed to train narrow nets with such width (due to the fundamental difficulty), and we indeed propose a new method to train narrow nets.
> > >
> > > We thank the reviewer again for the time spent reviewing and for the support. Hope the above response can better position our paper.
> > >
> > > [R1]: Bubeck, et al.. "Network size and weights size for memorization with two-layers neural networks."
> > >
> > > [R2]: Daniely, Amit. "Neural Networks Learning and Memorization with (almost) no Over-Parameterization." NeurIPS 2020. https://proceedings.neurips.cc/paper/2020/file/662a2e96162905620397b19c9d249781-Supplemental.pdf *

---

> > ### Public Comment · ~Bharath_B_N1 · 2023-08-27
> > **m missing in the Lipshitz constant calculation?**
> >
> > The Lipschitz constant calculation should have an order m term and looks like it is missing. If this is true, then the v should also scale down with m as 1/m. In this case, what is the guarantee that the global minima exists within the small ball?

---

### Official Review · Reviewer_hn5z · 2021-07-11

**Rating:** 6
**Confidence:** 4

**Summary:**

The authors study expressivity and trainability for one hidden layer neural networks. They showed that, when the number of neurons is larger than $n/2d$, where $n$ is the sample size and $d$ is the input dimension, then it has strong expressivity, i.e.,  with probability one, one global-min solution with zero training loss exists. The authors also propose a method to find such a global-min during training.

**Limitations And Societal Impact:**

Yes, the authors have adequately addressed the limitations and potential negative societal impact of their work.

**Main Review:**

Originality: The related works are adequately cited. The novelty of this paper is high. The study of the expressive power and trainability of neural networks is very important. The previous works on the expressivity of one hidden layer neural networks, such as [Soudry and Carmon 2016] and [Xie et al. 2017], did not discuss whether a stationary solution exists around the initial point. The results in this paper improve such previous works in a significant way. Therefore, I think this is a significant contribution to deep learning immunity. One disadvantage of this paper is that the authors only study one hidden layer neural networks, it will be more interesting to investigate multi-layer neural networks in the future.

Quality: This paper is technically sound.

Clarity: This paper is clearly written and well organized. I find it easy to follow.

Significance: I think the results in this paper is important, as explained above.

**Time Spent Reviewing:**

4 hours

---

> ### Author Response · Authors · 2021-08-10
> **To Reviewer hn5z**
>
> Thanks for your encouraging and supportive comments! Yes, the 'existence of stationary point around the initial point' serves as a foundation of the trainability, it is important but often ignored (especially for narrow nets). We are glad that the reviewer appreciates the importance of this topic.
>
> The extension to multi-layer networks is interesting but requires a lot more effort. The good news is that we already have some preliminary results on it (in Section 3.4). Analysis of multi-layer networks will be our major future direction.

---

### Official Review · Reviewer_Ep5m · 2021-07-13

**Rating:** 6
**Confidence:** 3

**Summary:**

Recent theoretical analysis on two-layer neural networks usually assumes the number of hidden neurons to be larger than the number of training samples so that the model could fit the data perfectly. This paper works on scenarios where the number of hidden neurons is smaller than the training samples. Clearly the small number of hidden neurons makes it harder to fit the data perfectly. Also some nice properties of the over-parameterized neural networks are missing, making it harder to prove guarantees on the optimization procedures.

The authors prove that for two layer neural networks even when the hidden neuron number is smaller than n it can still fit the data, demonstrating the expressivity of the two-layer network compared to the single layer network. The authors also propose a constrained optimization procedure with a new initialization method so that the global optimality is guaranteed.


**Limitations And Societal Impact:**

the authors adequately addressed the limitations and potential negative societal impact of their work

**Main Review:**

Analysis on limited width neural networks is in general scarce. Even though the analysis is still limited to two layer networks with quadratic loss, this work is making progress towards understanding the ``deep” models in terms of expressivity and optimum.

Just one question about the wording in Theorem 1: in line 130 should the ‘’for any small neighborhood” be ‘’there exists a small neighborhood” instead? I believe the Implicit Function Theorem is saying “ there exists a small neighborhood”.


**Time Spent Reviewing:**

3 hours

---

> ### Author Response · Authors · 2021-08-10
> **To Reviewer Ep5m**
>
> Thanks for your encouraging comments!
> You are right, the inverse function theorem(ITF) states 'there exists a neighborhood' (Can be seen in Appendix A.1). However, the 'existence of such a neighborhood' will imply 'IFT holds for any subset of this neighborhood', so actually, thm1 holds for 'any small (enough) neighborhood'. We will re-polish our statement and make it clearer in the full paper.

---

### Official Review · Reviewer_LkFg · 2021-07-14

**Rating:** 5
**Confidence:** 3

**Summary:**

This paper studies memorization capacity of depth-2 neural networks showing under certain conditions that 2n/d units suffice to perfectly memorize n dimensional labeled data points and that small gradient implies small (global loss).

**Limitations And Societal Impact:**

Yes

**Main Review:**

I find the description of the results and related work lacking. First of all the paper ignores recent results about memorization of relu networks that work with similar assumptions and parameters such as "Neural networks learning and memorization with (almost) no over-parameterization" and "Network size and weights size for memorization with two-layers neural networks". Do these existing results imply already the results in this paper? Second, the paper does not detail if points with small gradient can be approached efficiently (no running time analysis is given). I would suspect that if true this result would be hard to prove as current papers regarding convergence of over-parametrized ReLU's of depth 2 require far more parameters than 2n/d.
Perhaps the i.i.d assumption  can be used here. In general the paper cites 15-22 (dealing with the ReLU activation function) whereas the paper deals with smooth activation functions. I do not even know if the results of 15-22 hold for depth-2 networks with such activation functions. It would be good if the authors could comment on this point.

The paper is full of typos and the bibliography is not sorted according to alphabetical order.

**Time Spent Reviewing:**

7 hours.

---

> ### Author Response · Authors · 2021-08-10
> **To Reviewer LkFg**
>
> Thanks for your helpful feedback! Here are our responses to your concern.
> > *Q1: two related works [R1] [R2]*
>
> Thanks for pointing out these two references! These two papers are also mentioned by Reviewer wktj. To avoid repetition of the long reply, we suggest you refer to our reply 'To reviewer wktj: Part I: Related Works', where we provide a detailed comparison between these papers and ours. (you can find it by searching 'Part I: Related Works' on this webpage. )
>
> > *Q2: Running time analysis.  I would suspect that if true this result would be hard to prove as current papers regarding convergence of over-parametrized ReLU's of depth 2 require far more parameters than 2n/d. Perhaps the i.i.d assumption can be used here.*
>
> Thanks for pointing out this issue. Yes, the iteration complexity is important. This interesting topic will be left to future work. We have some progress on it and it requires much additional effort.
>
> As for "i.i.d. assumption", do you mean we can analyze iteration complexity under the extra assumption that data points are i.i.d. drawn from a certain data distribution (such as Gaussian or linearly separable data)? Thank you for this suggestion; we will think along this line.
>
> > *Q3: In general the paper cites 15-22 (dealing with the ReLU activation function) whereas the paper deals with smooth activation functions. I do not even know if the results of 15-22 hold for depth-2 networks with such activation functions. It would be good if the authors could comment on this point.*
>
> Thanks for your comments. From our understanding (correct us if wrong), you have a question on: 'it is not sure whether global convergence of GD is true for smooth activation, even for wide cases' . Our answer would be 'Yes, it is true, and it has been well studies exactly by some papers in citation 15-22'.  To clarify, Lee et al. [our citation 20], Jacot et al. [our citation 21], and Chizat et al. [our citation 22] deal with smooth activation, and the rest of citation 15-22 deal with ReLU. Despite the different activation & settings, they all prove the global convergence of GD for very wide networks. Therefore, the global convergence of GD is already proven to be true for a broad class of activation, **not just for ReLU** (hope it can address your concern).
>
> Let us further elaborate on why we choose smooth activation instead of ReLU (This may not be the real focus of your question. But for completeness, let us further explain the difference of activation in the 'narrow' case. )
>
> In our paper, we analyze smooth activation because we follow the approach of Li et al.'2018 [R1] to show the full-rankness of the Jacobian. Meanwhile, there exist papers that show the full-rankness of Jacobian for ReLU activation, e.g., Du et al.'18 [R2]. We choose [R1] since we find a simple way to extend their proof (they prove full-rankness of Jacobian of wide nets, and we extend the proof to prove the full-rankness of Jacobian of narrow nets). We suspect that combining some techniques of this paper and the proof of, say, Du et al.'18, can lead to results for narrow-nets that cover ReLU.
>
> In summary, as we pointed out above, the difference in activation only causes the difference in the 'technical level' of proving 'full-rankness of Jacobian', which does not change the 'essence' of the problem. The general proof idea of 'controlling the iterates' remains the same.
>
> [R1] Li, Dawei, Tian Ding, and Ruoyu Sun. "On the benefit of width for neural networks: Disappearance of bad basins." arXiv preprint arXiv:1812.11039 (2018).
>
> [R2] Du, Simon S., et al. "Gradient descent provably optimizes over-parameterized neural networks." arXiv preprint arXiv:1810.02054 (2018).
>
>
> > *Q4: The paper is full of typos and the bibliography is not sorted according to alphabetical order*
>
> Thanks for pointing out these issues. We will re-order the citation and correct the typos.

---

### Decision · Program_Chairs · 2021-09-28

**Decision:**

Accept (Poster)

**Comment:**

This paper makes a significant contribution and the results were found interesting and technically innovative.

However, the authors failed to discuss relevant literature in their original paper, in particular Daniely's and Bubeck et al. papers. Regarding Bubeck et al, this is an easily fixable mistake, and there seem to be no real issue here that cannot be resolved.

However, as far as Daniely's work, the relevance is too large to be brushed off. Now, the authors did make convincing arguments in their rebuttal. But, bottom-line, they make arguments about the validity of the result in Daniely. Now, of course this is completely acceptable, and from my brief view of their comments I am even inclined to believe that they have a point.  But such claims and discussions should have been part of the original paper and must go through the scrutiny of peer-review, they shouldn't be assessed in the limited form of the discussion period.

Given that Neurips does not accept revised versions, it is not possible to assess how the paper might look like once these issues will be resolved by the authors, and therefore I cannot recommend acceptance.

**Consistency Experiment:**

NeurIPS has a long history of experimentation. In 2014, NeurIPS ran an experiment in which 10% of submissions were reviewed by two independent committees to quantify the randomness in the review process. This year, we repeated a variant of this experiment to see how the quality of the review process has changed over time.  This paper was part of the experiment and was therefore assigned to two committees (consisting of reviewers, an Area Chair, and a Senior Area Chair) that reached independent decisions.  If both committees made the same recommendation, this recommendation was followed. If a single committee recommended acceptance, the paper was accepted (with the exception of a few cases in which the other committee identified what we considered a fatal flaw, e.g., an error in a key result).

This copy’s committee reached the following decision: **Reject**

The other committee assigned to the paper recommended **Accept (Poster)**.  You can find the other set of reviews, along with any follow up discussion with the authors here:
https://openreview.net/forum?id=ZBYphQE_hgp

**The paper will be accepted conditioned on the revision appropriately addressing the concern described in the meta-review.  The paper must pass a re-examination to ensure that the claims made with respect to the work of Daniely are correct.**